



# Measurement on PM and its chemical compositions for real-world emissions from non-road and on-road diesel vehicles

Min Cui[a], Yingjun Chen[a*], Cheng Li[b], Junyu Zheng[b*], Chongguo Tian[c], Caiqing Yan[d],

Mei Zheng[d]

[a] Key Laboratory of Cities' Mitigation and Adaptation to Climate Change in Shanghai (China Meteorological Administration), College of Environmental Science and Engineering, Tongji University, Shanghai, China

[b] Laboratory for Atmospheric Research and Environmental Simulation, School of

Environment and Energy, South China University of Technology, Guangzhou, China

[c] Key Laboratory of Coastal Zone Environmental Processes and Ecological Remediation, Yantai Institute of Coastal Zone Research, Chinese Academy of Sciences, Yantai, China

[d] State Key Joint Laboratory of Environmental Simulation and Pollution Control,

College of Environmental Sciences and Engineering, Peking University, Beijing, China

*Correspondence to*: Yingjun Chen (yjchentj@tongji.edu.cn)

Junyu Zheng (Zhengjunyu_work@hotmail.com)

**Abstract.** With increasing population of both non-road and on-road diesel vehicles,

the adverse effects of PM and its compositions (such as elemental carbon (EC), polycyclic aromatic hydrocarbons (PAHs)) on air quality and human health get more and more attention. However, characteristics of PM and its compositions emitted from diesel vehicles, particularly measured under real-world condition, are scarce. In this study, six excavators and five trucks, involving wide-range emission standards and

working in different operating modes, were tested to characterize constituents of PM (including organic carbon (OC), EC, water soluble ions (WSIs), elements, and organic species such as PAHs, n-alkanes, hopanes and steranes). The average emission factors



of PM ($EF_{PM}$) for excavators and trucks were 829 $\pm$ 806 and 498 $\pm$ 234 mg kg$^{-1}$ fuel, respectively, which are comparable with other studies. However, $EF_{PM}$ was significantly affected by fuel quality, operating modes and emission standards. High correlation ($R^2$=0.79, p<0.01) existed between the $EF_{PM}$ for excavators and the sulfur

contents in fuel. The highest average $EF_{PM}$ under working mode for excavators was 904 $\pm$ 979 mg kg$^{-1}$ fuel due to high engine load under this mode. From pre-stage 1 to stage 2 excavators, the average $EF_{PM}$ for excavators with different emission standards decreased by 58 %. Similarly, for trucks the average $EF_{PM}$ under non-highway condition (548 $\pm$ 311 mg kg$^{-1}$ fuel) was higher than those under highway condition

(497 $\pm$ 231 mg kg$^{-1}$ fuel). Meanwhile, reductions from China II and China III to China IV trucks were 63.5 % and 65.6 %, respectively. Generally, PM compositions emitted from excavators dominated by OC (39.2 % $\pm$ 21.0 %), EC (33.3 % $\pm$ 25.9 %), and while for trucks, PM dominated by EC (26.9 % $\pm$ 20.8 %), OC (9.89 % $\pm$ 12 %) and WSIs (4.67 % $\pm$ 5.74 %). Several differences of compositions were observed among

various operating modes, emission standards and fuel quality. The average OC/EC ratios under idling and working modes for excavators were 3 and 4 times higher than those in moving modes. Although $EF_{PM}$ for excavators and trucks reduced with stringent emission standards, the fractions of elements for excavators ranged from 0.49 % to 3.03 % from pre-stage 1 to stage 2, and fraction of WSIs for China IV truck

was 6 fold higher than those from other trucks. Furthermore, compared with the results from other diesel vehicles, wide ranges of ratios of BaA/(BaA+Chry) (0.26-0.86), IcdP/(IcdP+BghiP) (0.20-1.0) and Flua/(Flua+Pry) (0.24-0.87) for excavators were found, which may be attributed to the complex of operating modes for excavators. Although fractions of total 16 PAHs for excavators and trucks were

similar, the total of BaPeq that was used to evaluate the carcinogenic risk was 31 times higher than those for trucks. Therefore, more attention should be paid to non-road vehicle's emission.

**Keywords**

Diesel vehicles; excavators; trucks; PM; chemical composition; impact factors



**Copyright statement**

We confirm that the material is original and has not been submitted elsewhere.

## 1   Introduction

Particulate matter emitted from diesel vehicles have significantly adverse impacts on air pollution, human health and global climate change, and therefore should be examined closely (Aggarwal et al., 2015, 2016). Many studies have reported that diesel vehicles exhaust is a major source of emissions in ambient PM (Oanh et al., 2010, Zhang et al., 2015b). For example, it is reported that vehicle exhaust contributed to almost 30 % of emissions in ambient PM in 9 cities in China in 2015 (MEP 2016). The International Agency for Research on Cancer (IARC) found that exposure to diesel exhaust causes lung cancer (IARC 2012). It is reported that more than 25 million children breathe polluted air on diesel school buses, causing adverse respiratory health (Adar et al., 2015). EC emission, a major contributor to current global warming and human health, accounts for nearly 34 % from off-road diesel vehicles in the USA (USEPA 2015).

The populations of on-road and non-road diesel vehicles have increased considerably in China, especially for non-road diesel vehicles, causing severe emission situation. According to reports, the number of on-road diesel vehicles increased from 11.0 million in 2009 to 32.8 million in 2015, and the number of non-road diesel vehicles increased from 20.6 million in 2006 to 33.6 million in 2012 (CCCMIY et al., 2013, MEP 2016). According to China vehicle environmental management annual report (MEP 2016), 0.56 million tons of PM were emitted from on-road mobile sources and almost higher than 90 % of PM came from on-road diesel vehicles emission in 2015. Wang et al. (2016) estimated the emission inventory from non-road equipment (including agricultural equipment, river/ocean-going vessels, locomotives, and commercial airplanes) and found that there are 349 thousand tons PM emission from non-road vehicles in China in 2012. As an important type of non-road diesel vehicle, the number of construction instruments increased from 1.97 million to 5.85 million during 2006 to 2012 in China (CCCMIY 2013). According to





Zhang et al. (2010) Pearl River Delta (PRD) region's PM emission from construction instruments significantly accounted for 26.5 % of the total non-road vehicles in 2006. As one of the most abundant types of construction instruments, diesel consumption and PM emission from excavators were 7450 and 34.8 thousand tons in 2007 (Li et al., 2012).

In order to control diesel vehicles PM emission pollution, China has started to implement emission standards early in 2001 for light-duty diesel vehicles and heavy-duty diesel vehicles (SEPA et al., 2001). Those standards have been tightened from China I to China V in 12 years. Although emission standards for on-road diesel vehicles were formulated to China V, insufficient diesel fuel quality retards implementing of emission standards (Yue et al., 2015). The China IV emission standards for on-road diesel vehicles are not fully implemented until now. Compared with on-road diesel vehicles, the implementing time for emission standards for non-road diesel vehicles lagged behind. China has implemented two emission standards for new non-road diesel engines, stage 1 and stage 2, in 2007 and 2009, respectively. Furthermore, this first implemented time in China was 7 years later compared with the USA (USEPA 2003, SEPA et al., 2007).

The fundamental work of $EF_{PM}$ that is important parameter in the compilation of emission inventory for on-road and non-road diesel vehicles in China is relatively weak and uncertain (Huang et al., 2011). Recently, most of $EF_{PM}$ used in emission inventory research came from developed countries. Wang et al. (2016) estimated emission inventory from non-road equipment and suggested that real-world measurements of emissions for non-road equipment are desperately needed. Along with increasingly serious environmental problems, PM emission from on-road diesel vehicles has been taken seriously in China. There are considerable studies about on-road tests to study PM from on-road diesel vehicles. Wu et al. (2015) tested 17 in-use diesel trucks in Beijing using a portable emission measurement system (PEMS) and calculated the $EF_{PM}$ of those vehicles. Moreover, Zhang et al. (2015b) measured the real world PM emission factors from in-use HDDTs using PEMS. In addition, $EF_{PM}$ emitted from non-road diesel vehicles on real-world conditions was scarce in



China. To our knowledge, dynamometer test was used by most of studies to research non-road vehicle emission (Liang et al., 2005, Liu et al., 2015, Pietikainen et al., 2015). Liu et al. (2015) measured the PAH and nitro-PAH emission from non-road diesel engine, which was conducted utilizing the dynamometer test cycles required by U.S. EPA Tier 4 Final standards. Fu et al. (2012) tested 12 excavators using PEMS to determine PM emission factors under different working modes. However, due to the complexity of real-world conditions, such as lagging diesel quality and changing emission standards, the on-board measurements need to be expanded to improve localization of $EF_{PM}$ for on-road and non-road diesel vehicles in China as soon as possible.

Chemical constituents are important for studies of source apportionment, human health and climate change. Primary PM emitted from diesel vehicles contains variety of chemical compositions, such as OC, EC, water soluble ions, elements, and organic species (n-alkanes, PAHs, hopane and sterane). Several previous field studies have focused on chemical compositions of PM emitted from diesel vehicles. Zhang et al. (2015b) characterized PM compositions (OC, EC, WSIs and elements) emission from HDDTs. Wu et al. (2016) reported the detailed chemical composition of $PM_{2.5}$, including the organic carbon (OC), elemental carbon (EC), water soluble ions (WSIs), and element contents, emitted from China III and China IV diesel trucks. In 2012, Fu et al. (2012) tested 12 excavators to determine optical-based PM emission factors, which was the first on-board test for excavators in China. However, the specific characteristics of PM and compositions emitted from diesel vehicles, especially for organic matters, are lacking.

In this study, PM and its composition emitted from on-road and non-road diesel vehicles were measured in order to (I) test emission factors of PM for excavators and trucks under real world conditions; (II) identify impact factors of PM and its compositions for non- and on-road diesel vehicles; and (III) characterize chemical components of PM from excavators and trucks. Our results of this study could provide valuable information for development of effective control policies and reduce PM emission from excavators and trucks.



## 2 Experimental

### 2.1 Diesel vehicles and operation modes selection

In this study, six excavators and five trucks were selected to cover a range of emission standards, manufacturers and engine loads. The detailed information of the excavators and trucks is shown in Table 1. The tested excavators were divided into two groups based on their emission standards: three pre-stage 1 excavators and three stage 2 excavators. As shown in Figure S1, the annual productions of excavators have not changed much in 2007-2009, during which stage 1 non-road vehicle emission standards was implemented, varying from 70 to 85 thousand pieces of excavators. Therefore, excavators conducted with pre-stage 1 and stage 2 emission standards were chosen in this study. Based on China national standard (SEPASAQSIQ 2007), excavators can be divided into five types ($0<P<8$ kw; $8<P<18$ kw; $18<P<37$ kw; $37<P<75$ kw; $75<P<130$ kw; $130<P<560$ kw) according to the rated power (P). Thus, each type of excavators divided by emission standards included 0-75 kw, 75-130 kw and 130-560 kw excavators, which represent the low,medium and high power excavators, respectively. For reflecting the real operation environment, three operating modes for excavators were selected idling mode, moving mode and working mode, respectively. Further description of these three modes were listed in Fu et al. (2012). In this study, average duration consumption in idling, moving and working were 41.7, 24.0 and 28.5 minutes, respectively.

For diesel trucks, there were three types of trucks according to emission standards, one China II truck, three China III trucks and one China IV truck. Practicality, just China III trucks contained three trucks including light-duty, medium-duty and heavy-duty diesel trucks. Based on the traffic control measures and driving conditions of on-road diesel trucks, different predesigned routes were chosen for different emission standards and size trucks in Yantai, Shandong province in China (Figure 1). The routes chosen for China III and China IV light-duty trucks included arterial road (non-highway 1), secondary road (non-highway 2) and highway 1. The lengths of those roads were 19, 35 and 17 km, respectively. The route chosen for China II heavy-duty truck was special used for "yellow label car" (non-highway 3) which was





25 km. The routes chosen for China III medium-duty and heavy-duty trucks included non-highway 4 and highway 2. The lengths of those roads were 47 and 23 km, respectively. The detail velocity and road grade information for all of the tested routes were shown in Figures S2 and S3.

**2.2 On-board emission measurement system**

The on-board emission measurement system was self-designed and combined in our laboratory (Figure 2). The description of the used on-board emissions test system was given by our previous study (Zhang et al., 2015a). Briefly, this system has two main functional parts, including Photon II for flue gas (HC, CO, $CO_2$, $SO_2$, NOx) analyzer and PM sampler system. The PM sampler system consists of a dilution system, and five exhaust channels behind this dilution system. Two channels were connected with PM samplers, and the others were blocked. When sample the PM emitted from excavators, emission measurement system was put on a truck connected to the excavators exhaust via stainless steel pipe.

**2.3    Chemical analysis**

2.3.1.    Fuel quality analysis

Fuel quality has a great effect on PM emission from vehicles (Liang et al., 2005, Zhang et al., 2014, Cui et al., 2016). Due to poor fuel quality used in excavators, diesels for each of the tested excavators were collected and tested. The results for fuel quality analysis were shown in Table 2. Comparing the diesel quality used in this study with the diesel quality standards for non-road vehicles (GB 252-2015) (SEPA et al., 2015), it was found that most of the sulfur contents in diesel used in this study were higher than those allowed in GB 252-2015. Additionally, the sulfur content in the diesel used by E4 was 1100 ppm higher than those in diesel used for other excavators. Furthermore, the ash content of E4's diesel was 4.16%, about therefore 420 times higher than the limit value in GB 252-2015.

2.3.2. PM and chemical composition analysis

The quartz-fiber filters were weighed before and after sampling to determine mass concentrations of PM. Before each weighing, filters were balanced at 25 $^{\circ}$C and 40 % relative humidity for 24 h. The filters were weighed there times before and after





sampling to insure that the error for each measurement was as low as possible. WSIs were analyzed by ion chromatograph (Dionex ICS3000, Dionex Ltd., America) following the method of Cui et al. (2016). Elements was analyzed using inductively coupled plasma coupled with mass spectrometer (ICP-MS, ELAN DRC II type, Perkin Elmer Ltd., Hong Kong).

Because there was not enough organic matters on each filter, we combined different operating modes or routes filters for analysis for each diesel vehicles according to the proportion of tested time. Quartz filter samples spiked with internal standards (including acenaphthene-$d_{10}$, benzo[a]anthracene-$d_{12}$, pyrene-$d_{10}$, coronene-$d_{12}$, cholestane-$d_4$, $n$-C15-$d_{32}$, $n$-C20-$d_{42}$, $n$-C24-$d_{50}$, $n$-C30-$d_{58}$, $n$-C32-$d_{66}$, $n$-C36-$d_{74}$) were ultrasonically extracted two times in 30 ml of 1:1 mixture of hexane and dichloromethane for 10 min. All extracts for each sample were combined, filtered and concentrated to ~0.5 ml.

Organic species including n-alkanes, PAHs, hopane and sterane were analyzed using GC-MS (Agilent 7890A GC-5975C MS) equipped with a DB-5MS column (length 30 m ×i.d. 0.25 mm × thickness 0.25 μm). GC operating program was as following: 60 $^o$C with static time of 4 min, 5 $^o$C min$^{-1}$ to 150 $^o$C with static time of 2 min, then ramped to 306 $^o$C at rate of 3 $^o$C min$^{-1}$ with a static time of 20 min; and GC conditions:injector temperature was 290 $^o$C, volume of injector was 2 μL, carrier gas was helium, flow rate of gas was 1.2 ml min$^{-1}$. The electron impact (EI) mode at 70 eV and selected-ion-monitoring (SIM) mode were selected to determining PAHs, hopane and sterane. For organic matters, the blank samples and recovery rate were measured. The blank concentrations were subtracted from sample concentrations.

Chemical constituents of PM analyzed in this study are listed as follows: OC, EC, WSIs: $SO_4^{2-}$, $NO_3^-$, $Cl^-$, $NH_4^+$; Elements: Na, Mg, K, Ca, Ti, V, Cr, Mn, Fe, Co, Ni, Cu, Zn, Pb); n-alkanes: C12-C40; sixteen priority PAHs: naphthalene (Nap), acenaphthylene (Acy), acenaphthene (Ace), fluorine (Flu), phenanthrene (Phe), anthracene (Ant), fluoranthene (Fluo), pyrene (Pyr), benzo [a]anthracene (BaA), chrysene (Chry), benzo[b]fluoranthene (BbF), benzo[k]fluoranthene (BkF), benzo[a]pyrene (BaP), indeno[1,2,3-cd]pyrene (IcdP), dibenz[a,h]anthracene (DahA)





and benzo[ghi]perylene (BghiP); Hopane and sterane: ABB-20R-C27-Cholestane (ABB), AAA-20S-C27-Cholestane (AAA), 17A(H)-22,29,30-Trisnorhopane (Tm), 17A(H)-21B(H)-30-Norhopane (30AB), 17A(H)-21B(H)-Hopane (29AB).

**2.4 Data processing**

2.4.1.   Fueled-based emission factors

Fueled-based emission factors were calculated by the carbon-mass balance formula.

$$EF_i = \frac{\Delta X_i}{\Delta CO_2} \cdot \frac{M_i}{M_{CO_2}} \cdot EF_{CO_2} \tag{1}$$

Where $EF_i$ and $EF_{CO_2}$ (g kg$^{-1}$ fuel) are the emission factors for species i and $CO_2$,

$\Delta X_i$ and $\Delta CO_2$ (mol m$^{-3}$) are the background-corrected concentrations of i and $CO_2$,

and $M_i$ and $M_{CO_2}$ (g mol$^{-1}$) represent the molecular weights of species i and $CO_2$.

$$EF_{CO_2} = R_{FG} \cdot c(CO_2) \cdot M_{CO_2} \tag{2}$$

Where $c(CO_2)$ (mol m$^{-3}$) is the molar concentration of $CO_2$, $R_{FG}$ (m$^3$ kg$^{-1}$ fuel) represents the flue gas emission rate.

$$R_{FG} = \frac{C_F}{c(C_{CO}) + c(C_{CO_2}) + c(C_{PM})} \tag{3}$$

Where $C_F$ (g C kg$^{-1}$ fuel) represents the mass of carbon in 1 kg diesel fuel, and $c(C_{CO})$, $c(C_{CO_2})$, $c(C_{PM})$ (g C m$^{-3}$) represent the mass concentrations of carbon as CO, $CO_2$, PM in the flue gas, respectively.

2.4.2.   Average fuel-based emission factors for excavators and trucks

The average fuel-based emission factor for each excavator under different operating modes calculated by follows:

$$EF_{i,j} = \sum EF_{i,j,g} \times P_{j,g} \tag{4}$$

Where $EF_{i,j}$ (g kg$^{-1}$ fuel) is the average emission factor of species i for excavator j, $EF_{i,j,g}$ (g kg$^{-1}$ fuel) represents emission factor of species i for excavator j under g mode, and $P_{j,g}$ (%) is proportion of activity time (Fu et al., 2012) for excavator j under g mode.

The average fuel-based emission factor for each truck under different driving conditions calculated by follows:





$$EF_{i,j} = \sum EF_{i,j,s} \times P_{j,s} \tag{5}$$

Where $EF_{i,j}$ (g kg$^{-1}$ fuel) is the average emission factor of species i for excavator j, $EF_{i,j,s}$ (g kg$^{-1}$ fuel) represents emission factor of species i for excavator j under s condition, and $P_{j,s}$ (%) is proportion of activity time for truck j under s condition .

2.4.3. Benzo[a]pyrene equivalent concentration (BaP$_{eq}$)

Because of different carcinogenic risks for each PAH, the BaP$_{eq}$ for parent PAHs were given. The BaPeq was calculated by multiplication of the measured concentrations by the respective potency equivalent factor (PEF) (Mirante et al., 2013). The PEF values were obtained from Wang et al. (2008).

## 3 Results and discussion
### 3.1 Particulate matter fuel-based emission factors for excavators

$EF_{PM}$ for excavators exhaust are presented in Figure 3, with the detailed information shown in Table S1. The maximum PM fuel-based emission factor was almost 37 times higher than the minimum under different operating modes for different vehicles. In general, the average $EF_{PM}$ for different excavators ranged from 96.5 to 2323 mg kg$^{-1}$ fuel, with an average of 829 $\pm$ 806 mg kg$^{-1}$ fuel. The $EF_{PM}$ values of excavators reported by Fu et al. (2012) were within the range of $EF_{PM}$ in this study but in a narrower range. The reason for the more widely ranged $EF_{PM}$ in this study may be that the selection of excavators. The excavators selected by Fu et al. included stage 1 and stage 2 emission standards, while this study tested excavators pre-stage 1 and stage 2 emission standards. Therefore, the range of $EF_{PM}$ in this study may reflect the general excavator's PM emission situation in China.

$EF_{PM}$ could be affected by many factors. Some variation characteristics about the $EF_{PM}$ values due to the different fuel quality, emission standards and operating modes were summarized as follows. Firstly, fuel quality has great impact on $EF_{PM}$ for excavators. As shown from Figure 3, high correlation ($R^2$=0.79, P<0.01) was found between the average emission factors for excavators and the sulfur content in fuel, which is consistent with the results studied from Yu et al. (2007). Secondly, $EF_{PM}$ decreased with enhancing of emission standards for excavators. The measured $EF_{PM}$



for pre-stage 1 excavators under idling, moving and working conditions were 914 $\pm$ 393 mg kg$^{-1}$ fuel, 609 $\pm$ 38 mg kg$^{-1}$ fuel and 1258 $\pm$ 1295 mg kg$^{-1}$ fuel, respectively. The EF$_{PM}$ for stage 2 excavators under idling, moving and working conditions were 243 $\pm$ 236 mg kg$^{-1}$ fuel, 165 $\pm$ 144 mg kg$^{-1}$ fuel and 551 $\pm$ 587 mg kg$^{-1}$ fuel, respectively. Compared to pre-stage 1, EF$_{PM}$ of stage 2 excavators reduced 73 %, 73 % and 56 % in idling, moving and working modes, respectively. The average EF$_{PM}$ for excavators of different emission standards decreased 58 % from pre-stage 1 to stage 2, suggesting the effectiveness of emission control policy. Lastly, EF$_{PM}$ varied sharply between different operating modes for various excavators. Specifically, excavators under working modes have the highest EF$_{PM}$, which is higher than the value for other operating modes by more than 1 fold. The average EF$_{PM}$ for excavators under different driving conditions were 578 $\pm$ 467 mg kg$^{-1}$ fuel (idling), 343 $\pm$ 264 mg kg$^{-1}$ fuel (moving) and 904 $\pm$ 979 mg kg$^{-1}$ fuel (working), respectively. The highest average EF$_{PM}$ under working mode might be attributed to higher engine load, which causes lower air-fuel ratios and then prompted the PM production.

### 3.2 Particulate matter fueled-based emission factors for trucks

The EF$_{PM}$ for all measured trucks under different driving patterns varied from 176 mg kg$^{-1}$ fuel to 951 mg kg$^{-1}$ fuel. There were just tripled in PM emission factors for trucks from maximum to minimum. The average EF$_{PM}$ for tested diesel trucks was 498 $\pm$ 234 mg kg$^{-1}$ fuel. In comparison, Wu et al. (2016) reported an average EF$_{PM}$ for diesel trucks of 427 (95.6-1147 mg kg$^{-1}$ fuel) mg kg$^{-1}$ fuel and it is within the same range as our results.

Besides, The average EF$_{PM}$ of diesel trucks for different emission standards, vehicle sizes and driving patterns were provided under real-world conditions (Figure 4). The measured EF$_{PM}$ for China II, China III and China IV diesel trucks varied from 200 mg kg$^{-1}$ fuel to 548 mg kg$^{-1}$ fuel. EF$_{PM}$ for China II truck measured in this study was lower than the results obtained from Liu et al. (910-2100 mg kg$^{-1}$ fuel) (2009). The average EF$_{PM}$ for light-duty, medium-duty and heavy-duty diesel trucks were 524 $\pm$ 457 mg kg$^{-1}$ fuel, 459 mg kg$^{-1}$ fuel and 492 mg kg$^{-1}$ fuel, respectively. The average EF$_{PM}$ of trucks under non-highway and highway driving patters were 548 $\pm$ 311



mg kg$^{-1}$ fuel and 497 $\pm$ 231 mg kg$^{-1}$ fuel, respectively. As Figure 4 shows, reductions of EF$_{PM}$ for China II truck to China IV truck and from China III truck to China IV truck in EF$_{PM}$ were 63.5 % and 65.6 %, which indicated that improvements of emission standards for diesel trucks significantly decreased PM emission. It should be

noticed that EF$_{PM}$ for China III and light-duty diesel trucks were higher than other corresponding trucks. The reason may be attributed to poor driving conditions that include low average speed and more volatile in speed for those trucks (Figure S2 and Figure S3). Same tendency could be seen from Figure 4 that diesel trucks emitted more PM under non-highway condition (average speed: 28.5 km h$^{-1}$) than those under

highway condition (average speed: 60.7 km h$^{-1}$). Furthermore, the road grade was an another aspect effected the EF$_{PM}$ of on-road diesel trucks. For example, EF$_{PM}$ for T5 under highway road was lower than those for T1 because of bigger road grade for T5 under highway road than those for T1 (Figure S3).

### 3.3 Particulate matter composition for individual diesel vehicle

Four constituents were considered for reconstituting PM mass, in this study: (1) organic matter, which was estimated by multiplying the corrected OC by a factor of 1.6 (Almeida et al., 2006); (2) EC; (3) water soluble ions; (4) elements. The reconstituted mass for each excavator sampler was 74.7 %-123 % of measured mass, while reconstituted mass for diesel truck sampler was only 43.2 %-54.4 % of

measured mass (Figure 5). Except for uncalculated components, this discrepancy may be attributed to uncertainties in the weighing process (Dai et al., 2015).

3.3.1. Particulate matter composition for individual excavator

The chemical composition of PM for each excavator was shown in Figure 5 and Table S2. For each excavator, carbon component (OM+EC) was the dominant species,

consisted with previous study from a non-road diesel generator that had found the proportions of organic and element carbon in PM ranged from 70.2 % to 90.6 % (Liu et al., 2005). Because OC/EC ratio is also used to identify the source of atmospheric particulate pollution, deeper discussion about OC/EC ratios under different operating modes for each excavator was conducted (Figure 6). The average OC/EC ratios for

idling, moving and working modes were 1.57, 0.57 and 2.38, respectively. The





OC/EC ratio under idling was higher than 1 because soot generated at low temperature hardly and fuel-rich zone, which is similar to the research done by Liu et al. (2005). Furthermore, Liu et al. (2005) reported that the OC/EC ratios decreased with increasing non-road engines load. However, this trend couldn't be observed in this study. The OC/EC ratio was 2.38 under working mode, increasing again with load increasing, which consistent with the results from Zhang et al. (2014). Large OC/EC ratios differences for excavators under different operating modes were seen in Figure 6, which may be caused by a number of factors (such as transient working conditions, diesel sulfur content and extensive sources for OC) (Cocker et al., 2004, Liu et al., 2005).

As shown from Figure 5, WISs and elements fractions ranged from 0.335 % to 1.21 % and from 0.163 % to 7.50 % for all excavators. The total sum proportion of WISs and elements to PM was highest in excavator E6, followed by excavator E1. Generally, total sum proportion of WISs and elements to PM in excavator E1 was 4 to 14 times higher than corresponding proportions in other excavators. Sulfate and nitrate were the main WISs (79.1 %-90.0 % of WISs) for almost all excavators while the proportion of Cl⁻ of WISs for excavator E1 (67.2 %) was highest (Table S2). Fe, Ca, Na, Mg and K were relatively dominant in elements, but for E4 excavator, Fe, Zn and Cu were the most abundant elements. It may be attributed to that Zn is known from oil additives and Cu usually emitted from wear debris (Lin et al., 2015, Wu et al., 2015). Table 1 and Table 2 showed that excavator E4 produced in 2004 and the diesel quality used was poor, resulting in high Zn and Cu emission. Besides, the elements fractions for two excavators produced in 2013 (E1 (1.42 %) and E6 (7.50 %)) were higher than other excavators, which may indicate that elements emission was deteriorating and more stringent control technology should be developed to avoid the total elements adverse health effects.

In addition, the n-alkanes, PAHs, hopane and steranes fractions ranged from 3.6 % to 9.6 %, from 0.03 % to 0.24 % and from 0.001 % to 0.09 % for excavators, respectively. Liang et al. (2005) characterized diesel particulate matter emitted from non-road engine using dynamometer test and found that n-alkanes accounted for 0.83%



of PM, which was lower than the results obtained from this study. The main reasons are the low sulfur diesel fuel used in Liang's study and different methods used in obtained the PM. Contrary to what was observed for WISs and elements, Figure 5 showed that n-alkanes, hopane and steranes fractions were highest in excavator E4

while PAHs fraction was highest in excavator E3. It was said by Rogge et al. (1993) that n-alkanes, PAHs, hopane and steranes are mostly derived from incomplete combustion of fuel and lubricant oil.  By comparing the differences between fuel quality and performance of excavators, it could be deduced that n-alkanes, hopane and steranes were influenced by fuel quality and PAHs was affected by combustion

conditions in this study. PAHs isomer ratios have been widely used to distinguish the source apportionment in environmental receptors (such as sediments) (Liu et al., 2012). Yunker et al. (2002) found that the ratios of the principal mass 178, 202, 228 and 276 parent PAHs have a best potential to distinguish natural and anthropogenic sources. For excavators, the ranges of ratios of BaA/(BaA+Chry), IcdP/(IcdP+BghiP)

and Flua/(Flua+Pry) were 0.26-0.86, 0.20-1.0 and 0.24-0.87, with average of 0.47±0.27, 0.44±0.38 and 0.48±0.27, respectively (Figure 7). The average ratios of PAHs for excavators obtained in this study were similar with that from Liu et al. (2015) reported for non-road diesel engines. E4 excavators showed an obvious difference ratios of BaA/(BaA+Chry), IcdP/(IcdP+BghiP) and Flua/(Flua+Pry)

between the other excavators tested in this study. The isomer ratios of BaA/(BaA+Chry), IcdP/(IcdP+BghiP) and Flua/(Flua+Pry) for E4 were 0.86, 1.0 and 0.87, respectively, and it were different with the ranges for fuel combustion defined by Yuners et al. (2002). The ratios of PAHs emitted from diesel vehicles reported by Yuners et al. mainly refer to those from on-road diesel vehicles. However, operating

modes and fuel quality for non-road diesel vehicles are more complicated than those from on-road diesel vehicles. Therefore, the results in this study could give references for isomer ratios of PAHs for non-road diesel vehicles.

3.3.2 Particulate matter composition for individual diesel trucks

For diesel trucks, total carbonaceous composition (OM+EC) were accounted for

44.0 % (E1), 27.9 % (E2), 43.9 % (E3), 51.6 % (E4) and 46.3 % (E5) of PM, which is



lower than those reported in previous studies (Chow et al., 2011, Wu et al., 2015). The reason may be mainly attributed to the detection methods for organic carbon and elements carbon. Cheng et al. (2011) collected 333 $PM_{2.5}$ samples and analyzed OC and EC by two common thermal-optical methods (NIOSH and IMPROVE) and found

that NIOSH-defined EC was lower (up to 80 %) than that defined by IMPROVE. The thermal-optical method used in this study was IMPROVE, which would make content of OC under evaluated. Almost all OC/EC ratios for diesel trucks under different driving conditions calculated in this study were lower than 1, which was consistent with the conclusion from previous studies (Figure 6), except for the T2 and T4 trucks.

The OC/EC ratios for T2 under highway and non-highway driving conditions were 5.64 and 15.5, respectively. This result may be attributed to emission standard for T2 (China IV). Alves et al. (2015b) reported that modern diesel passenger cars (Euro 4 and Euro 5) exhibit high OC/EC ratios. Furthermore, OC/EC ratio for T4 under non-highway condition was 4.10, which may be caused by driving speed. Cheng et al.

(2015) reported that the OC/EC ratios were substantially above unity at idling and low load. As shown from Figure S2, the driving speed for T4 was zero in 500 seconds before driving.

    Sum of WISs and elements fractions were lower than 5 % for almost all of the diesel trucks except for T2 truck, consistent with the results gained from Zhang et al.

(2015b). $SO_4^{2-}$ was the most abundant ions for trucks T2 and T5, while $NO_3$ was the most abundant ions for trucks T1, T3 and T4. For T2 diesel truck, WISs (13.8 %) was the most significant component of PM after OC and it was higher by a factor of 4 to 10 than those in other trucks (Table S2). T2 truck is a China IV diesel vehicle and well-controlled combustion conditions caused more water emission, which accelerate

the translation from gas to WISs (such as $SO_2$ translate to $SO_4^{2-}$). As we can see from Table S2, Fe was the most abundant element for trucks T1, T3 and T5, while Ca was the most abundant element for trucks T1, T3 and T4. Compared with elements fractions in T2 (China IV) and T1 (China III) trucks, fractions changed from 0.09 % (T2) to 1.5 % (T1). Although the PM emission factors decreased with emission

standards, the WISs and elements contents increasing along with promoting the





emission standards for diesel trucks. In consideration of acid rain causing by sulfate and nitrate and adverse health effect caused by elements, great attention should be pay to this phenomenon.

The n-alkanes, PAHs, hopane and steranes fractions ranged from 0.85 % to 4.78 %, from 0.01 % to 0.54 % and from 0.002 % to 0.024 % for trucks, respectively. As shown in Table S2, C20 was the most abundant species in n-alkanes for truck T1, T2 and T4, while C19 was the most abundant species for truck T3 and T5. For PAHs, the most notable species was Pyrene, which was substantially higher than all other PAHs for all trucks. The proportions of n-alkanes, PAHs, hopane and steranes to PM were highest for truck T3, which may be affected by many factors, such as differences in engine rate power, complex reaction in the engine and driving conditions. As shown from Figure 7, scatters of isomer ratios for diesel trucks were covered from 0.28 to 0.35 for BaA/(BaA+Chry), from 0.08 to 0.22 for IcdP/(IcdP+BghiP) and from 0.08 to 0.39 for Flus/(Flua+Pry), with averages of $0.31 \pm 0.03$, $0.15 \pm 0.06$ and $0.23 \pm 0.12$, respectively. There were similar to the results from Schauer et al. (1999).

### 3.4 Average chemical constituent of PM emitted from diesel vehicles

3.4.1.  Average chemical constituent of PM for excavators

The average chemical component of PM for excavators was listed in Table 3. It appeared that carbonaceous was the dominant component and account for 72.5 % of the PM for excavators, whereas OC was the most abundant species (39.2 %). Total element was the second largest group and contributed 1.76 % of PM. For elements, the emission was obviously dominated by Fe which accounted for 46.3 % of the elements. In addition, Table 3 showed that the proportion of n-alkanes in PM for excavators (5.14 %) was higher than those for other organic matters (PAHs: 0.098 %; hopane and sterane: 0.026 %), and C20 and C19 were maximum carbon in n-alkanes. For the parent PAHs, the emissions were dominated by Pry and Fluo, followed by Nap and Chry.

To compare our results with other studies, Table 3 summarizes average source profile of PM for excavators derived in this study as well as those previously reported by others. As shown in Table 3, average fraction of total carbonaceous for excavators





tested in this study are consistent with those measured from marine engine, while the fraction of elements was lower than that for marine engine (Sippula et al., 2014). It is said that oxidation of soot was enhanced during increasing of transition metals for diesel engines (Kasper et al., 1999). The EC fraction of PM for excavators was higher

than those from Sippula et al. (2014), which may be attributed to lower metal fraction in excavators. Comparing results from this study with other references showed that the proportion of n-alkanes measured in this study is significantly higher than those emitted from marine engine (4 fold) and non-road generators (6 fold) (Liang et al., 2005). The reason may be attributed to different contents of aliphatic compounds

existing in diesel fuel used for those non-road vehicles (Sippula et al., 2014). For the marine engine and non-road generators, C22 and C17 were the most abundant specie in n-alkanes, respectively. PAHs emission was dominant by Phe for marine engine and Fluo for non-generators, which was different with the result obtained from excavators. This could indicate that the PM emitted from different types of non-road

diesel vehicles has various source profiles because of diverse operation conditions.

3.4.2. Average source profile of PM for trucks

As shown in Table 3, average emission of PM from trucks was dominated by carbonaceous (36.8 %), and followed by WISs (4.67 %) and elements (0.941 %). For individual species, sulfate and nitrate were the most abundant in water soluble ions,

and Fe was dominated in elements. Moreover, for organic matters, the average proportion of n-alkanes, PAHs, hopanes and steranes was 1.73 %, 0.130 % and 0.011 %, respectively. C20 was maximum carbon in n-alkanes and the emission of PAHs was dominated by Pry.

In comparison, emission of total carbon from this study was lower than previous

studies, whereas, WISs and elements fractions were relatively higher than results obtained from other research groups (Schauer et al., 1999, Alves et al., 2015a, Cui et al., 2016, Wu et al., 2016). There are several reasons could be used to explain the results, including fuel quality, driving condition, parameters of engines (fuel injection timing, compression ratio, fuel injector design) and experiment methods (Sarvi et al.,

2008b, Sarvi et al., 2008a, Sarvi et al., 2009, Sarvi et al., 2010). As shown in Table 3,





Fe was dominant in elements from results measured by on-road test and tunnels, which was similar with our results, while Zn and Na were dominant in elements from results obtained by dynamometer. Therefore, on-road test and tunnels measured in real world would reflect real PM emission better. For organic matters, the proportion

of PAHs, hopane and sterane to PM were consistent with the results from Schauer et al. (1999) and Cui et al. (2016). Similar with this study, the maximum carbon in n-alkanes was C20 measured by Schauer et al, and Pyr was most abundant species in PAHs reported by Cui et al. Thus, average profile of PM for on-road diesel trucks was relatively stable.

**3.5 Comparing average composition of PM for excavators with those from trucks**

Compared with the average $EF_{PM}$ of excavators and diesel trucks obtained in this study, average $EF_{PM}$ for excavators (836 $\pm$ 801 mg $kg^{-1}$ fuel) was higher than those for diesel trucks (498 $\pm$ 234 mg $kg^{-1}$ fuel). The result was understandable because state operations for excavators were more transient than those for trucks. Sarvi et al. (2010)

reported that particulate matter emission from diesel engines was typically low during steady state operation. Although average $EF_{PM}$ of excavators higher than that emitted from trucks, average $EF_{PM}$ of stage 2 excavator was 477 mg $kg^{-1}$ fuel, which was lower than that emitted by China II and China III trucks. Thus, appropriate regulations formulated for non-road diesel vehicles could improve the PM emission situation.

When we compared the average percentages of chemical components in PM for excavators with those for trucks, we found that there were some differences between excavators and trucks. In general, carbonaceous composition (95.9 %) and elements (1.76 %) fractions for excavators were higher than those for diesel trucks (42.8 % for carbonaceous composition and 0.94 % for elements). As shown in Figure 8, the $BaP_{eq}$

levels for excavators and trucks were absolutely difference, although the PAHs average percentages of PM were consistent between excavators and trucks. Almost all of the parent PAHs's $BaP_{eq}$ calculated in this study for trucks and excavators were higher than the datum from WHO that concentration caused 1/10000 risk of carcinogenic. Moreover, total average of BaPeq for excavators was 31 fold of those

for diesel trucks. Due to some adverse environmental effects and health hazards for



carbonaceous composition, elements and PAHs, the PM emission from excavators should be controlled urgently.

## 4 Conclusions

This study reports the characteristics of PM source profiles for excavators and trucks. Above all, PM emission factors emitted from excavators and trucks under different operating modes, emission standards and road conditions were obtained. The $EF_{PM}$ for different excavators ranged from 96.5 to 2323 mg $kg^{-1}$ fuel, with an average of 810 mg $kg^{-1}$ fuel and a high correlation ($R^2$=0.79, P<0.01) with the sulfur contents in the fuel. The highest average $EF_{PM}$ in working mode (904 ± 979 mg $kg^{-1}$ fuel) might be attributed to higher engine load, which caused lower air-fuel ratios. The average $EF_{PM}$ for tested diesel trucks of different emission standards and vehicle sizes under different driving conditions was 498 ± 234 mg $kg^{-1}$ fuel. The average $EF_{PM}$ for different emission standards excavators decreased 58 % from pre-stage 1 to stage 2. Moreover, reductions from China II truck to China IV truck and from China III truck to China IV truck in $EF_{PM}$ were 63.5 % and 65.6 %, respectively. Those indicate that improvements of emission standards for diesel trucks and excavators have decreased PM emission significantly. It should be noticed that $EF_{PM}$ for China III and light-duty diesel trucks were higher than those for other corresponding trucks. The reasons may be attributed to poor driving conditions that include low average speed and more volatile in speed for those trucks. For each excavator, carbon component (OM+EC) was the dominant species and accounted for approximately 74.1 %-123 % of PM. The average ranges of WISs, elements, n-alkanes, PAHs, hopane and sterane fractions for each excavator were 0.335 %-1.21 %, 0.163 %-7.50 %, 3.6 %-9.6 %, 0.03 %-0.24 % and 0.001 %-0.09 %, respectively. In contrast to other excavators, Zn and Cu were the second and third most abundant elements in excavators E4, which may be attributed to poor fuel quality and the old vehicles age. Besides, the elements fractions for two excavators produced in 2013 (E1 (1.42 %) and E6 (7.50 %)) were higher than other excavators, which may indicate that elements emission was deteriorating and more stringent control technology should be developed to avoid the total elements adverse

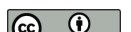



health effects. For excavators, the ranges of ratios of BaA/(BaA+Chry), IcdP/(IcdP+BghiP) and Flua/(Flua+Pry) were 0.26-0.86, 0.20-1.0 and 0.24-0.87, with average of $0.47 \pm 0.27$, $0.44 \pm 0.38$ and $0.48 \pm 0.27$, respectively. For diesel trucks, total carbonaceous composition (OM+EC) were accounted for 44.0 % (E1), 27.9 %

(E2), 43.9 % (E3), 51.6 % (E4) and 46.3 % (E5) of PM. For T2 diesel truck, wSIs (13.8 %) was the most significant component of PM after OC and it was higher than those in other trucks, within a factor of 4 to 10. The n-alkanes, PAHs, hopane and steranes fractions ranged from 0.85 % to 4.78 %, from 0.01 % to 0.54 % and from 0.002 % to 0.024 % for trucks. In comparison with other literatures, the characteristics

of average source profiles for different types of non-road diesel vehicles varied sharply, while for on-road diesel vehicles, those characteristics showed more stability. Although fractions of PAHs for excavators and trucks were identical, the total of BaPeq that was used to evaluate the carcinogenic risk was 31 fold of those for trucks.

Acknowledgements. This study was supported by the CAS Strategic Research Program (No.XDB05030303), the Natural Scientific Foundation of China (Nos.41273135 and 41473091), the Fundamental Research Funds for the Central Universities.

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



**Table captions**

Table 1 Specifications of tested excavators and trucks

Table 2 Diesel contents from excavators

Table 3 Comparison of average source profiles of PM for different diesel vehicles



Table 1 Specifications of tested excavators and trucks

| ID | manufacturers | Model years | Emission standards | Powers (kw) | Total weights (kg) | Displacements (L) | Working hours (h) | Mileages (km) |
|---|---|---|---|---|---|---|---|---|
| E1 | Volvo | 2013 | stage 2 | 169 | 30,500 | 7.1 | 2,751 | / |
| E2 | Hitachi | 2007 | pre-stage 1 | 162 | 30,200 | 9.8 | 16,166 | / |
| E3 | Sany | 2012 | stage 2 | 128 | 22,900 | / | 5,598 | / |
| E4 | Doosan | 2004 | pre-stage 1 | 110 | 22,000 | 8.1 | 12,000 | / |
| E5 | Doosan | 2007 | pre-stage 1 | 40 | 5,250 | 2.8 | / | / |
| E6 | Komatsu | 2013 | stage 2 | 35 | 5,300 | 2.4 | 780 | / |
| T1 | Futian | 2010 | China III | 68 | 4,495 | 2.6 | / | 100,238 |
| T2 | JAC | 2014 | China IV | 88 | 4,495 | 2.8 | / | / |
| T3 | Futian | 2011 | China III | 70 | 11,190 | 3.9 | / | 99,000 |
| T4 | Chunlan | 2002 | China II | 125 | 15,480 | / | / | / |
| T5 | JAC | 2011 | China III | 105 | 15,590 | 4.3 | / | 130,000 |





Table 2 Diesel contents from excavators

| ID | E1 | E2 | E3 | E4 | E5 | E6 | GB 252-2015 |
|---|---|---|---|---|---|---|---|
| Gross thermal value (MJ/kg) | 45.1 | 45.1 | 45.3 | 45.3 | 45.3 | 45.3 | / |
| Net thermal value (MJ/kg) | 42.4 | 42.4 | 42.7 | 42.8 | 42.6 | 42.5 | / |
| Kinematic viscosity (20 $^{\circ}$C)(mm$^2$/s) | 4.23 | 4.23 | 3.89 | 4.16 | 4.60 | 4.39 | 3.00-8.00 |
| Moisture (%) | n.d. | n.d. | n.d. | n.d. | n.d. | n.d. | / |
| Ash content (%) | 0.04 | 0.04 | 0.05 | 4.16 | 0.03 | 0.05 | 0-0.01 |
| C (%) | 86.3 | 86.3 | 86.4 | 86.8 | 85.9 | 85.9 | / |
| H (%) | 11.6 | 11.6 | 11.5 | 11.2 | 12.0 | 12.1 | / |
| O (%) | 1.99 | 1.99 | 2.01 | 1.85 | 2.07 | 1.86 | / |
| N (%) | 0.05 | 0.05 | 0.05 | 0.04 | 0.06 | 0.05 | / |
| S (ppm) | 400 | 400 | 700 | 1100 | 200 | 200 | <350 |





Table 3 Comparison of average chemical constituents of PM for different diesel vehicles (%)

| Vehicle types | Excavators | Trucks | Trucks | Medium-duty trucks | Diesel vehicles | Light-duty Diesel engines | Marine engine | Non-road generator |
|---|---|---|---|---|---|---|---|---|
| Methods | On-road | On-road | On-road | Dynamometer | Tunnel | Dynamometer | Dynamometer | Dynamometer |
| Reference | This study | This study | (Wu et al., 2016) | (Schauer et al., 1999) | (Cui et al., 2016) | (Alves et al., 2015b) | (Sippula et al., 2014) | (Liang et al., 2005) |
| **EC** | 33.3 | 26.9 | 55.3 | 30.8 | 39.5 | 69.9 | 14.1 | |
| **OC** | 39.2 | 9.89 | 31.8 | 19.7 | 27.2 | 12.7 | 60.0 | |
| **Ions** | 0.614 | 4.67 | 1.49 | 1.96 | 11.7 | 0.638 | | |
| $NH_4^+$ | 0.044 | 0.215 | 0.188 | 0.730 | 2.06 | 0.005 | | |
| $Cl^-$ | 0.098 | 0.110 | 0.247 | | 1.06 | 0.115 | | |
| $NO_3^-$ | 0.278 | 1.08 | 0.529 | 0.230 | 3.81 | 0.459 | | |
| $SO_4^{2-}$ | 0.193 | 3.27 | 0.529 | 1.00 | 4.80 | 0.059 | | |
| **Elements** | 1.76 | 0.941 | 0.493 | 0.200 | 12.8 | 0.069 | 3.17 | |
| Na | 0.245 | 0.047 | | | 0.287 | 0.041 | 0.564 | |
| Mg | 0.106 | 0.079 | | | 1.71 | 0.008 | 0.422 | |
| K | 0.197 | 0.028 | | | 0.872 | 0.002 | 0.671 | |
| Ca | 0.241 | 0.211 | | 0.030 | 5.69 | 0.017 | 1.01 | |
| Ti | 0.008 | 0.011 | 0.145 | | 0.206 | 0.0001 | 0.005 | |
| V | 0.001 | 0.000 | 0.001 | | 0.008 | | 0.044 | |
| Cr | 0.035 | 0.039 | 0.011 | 0.010 | 0.013 | | 0.010 | |
| Mn | 0.013 | 0.009 | 0.002 | 0.010 | 0.064 | | 0.006 | |



Continued Table 3

|          |        |        |        |       |       |        |        |
|----------|--------|--------|--------|-------|-------|--------|--------|
| Fe       | 0.815  | 0.276  | 0.247  | 0.050 | 3.71  | 0.0003 | 0.138  |
| Co       | 0.001  | 0.005  | 0.0002 | 0.010 | 0.002 |        | 0.006  |
| Ni       | 0.015  | 0.006  | 0.002  | nd    |       |        | 0.016  |
| Cu       | 0.042  | 0.107  | 0.004  | 0.010 | 0.013 |        | 0.130  |
| Zn       | 0.027  | 0.111  | 0.076  | 0.070 | 0.213 | 0.0001 | 0.130  |
| Pb       | 0.011  | 0.010  | 0.005  | 0.010 | 0.008 |        | 0.013  |
| **Alkanes** | 5.14 | 1.73 |       | 0.222 | 1.37  |        |        |
| C12      | 0.003  | 0.020  |        | 0.010 |       |        | 0.003  |
| C13      | 0.003  | nd     |        | nd    |       |        | 0.006  |
| C14      | 0.019  | 0.0003 |        |       |       |        | 0.020  |
| C15      | 0.057  | 0.013  |        | 0.001 |       |        | 0.056  |
| C16      | 0.201  | 0.062  |        | 0.005 |       |        | 0.116  |
| C17      | 0.107  | 0.144  |        | 0.003 |       |        | 0.265  |
| C18      | 0.587  | 0.215  |        | 0.002 |       | 0.049  | 0.148  |
| C19      | 0.777  | 0.308  |        | 0.002 |       | 0.120  | 0.126  |
| C20      | 0.977  | 0.311  |        | 0.052 |       | 0.260  | 0.074  |
| C21      | 0.516  | 0.290  |        | 0.022 |       |        | 0.014  |
| C22      | 0.769  | 0.143  |        | 0.028 |       | 0.264  | 0.001  |
| C23      | 0.349  | 0.099  |        | 0.025 |       | 0.177  | 0.001  |
| C24      | 0.245  | 0.061  |        | 0.022 |       | 0.128  | 0.001  |
| C25      | 0.197  | 0.032  |        | 0.014 |       | 0.083  | 0.0004 |





Continued Table 3

| | | | | |
|------|-------|---------|-------|--------|
| C26 | 0.119 | 0.016 | 0.019 | 0.075 |
| C27 | 0.031 | 0.009 | 0.014 | 0.056 |
| C28 | 0.023 | 0.004 | 0.011 | 0.058 |
| C29 | 0.013 | 0.002 | 0.003 | 0.046 |
| C30 | 0.007 | 0.001 | | 0.025 |
| C31 | 0.010 | 0.002 | | 0.017 |
| C32 | 0.010 | 0.001 | | 0.007 |
| C33 | 0.010 | 0.00001 | | 0.002 |
| C34 | 0.010 | 0.0004 | | |
| C35 | 0.013 | 0.00004 | | |
| C36 | 0.016 | nd | | |
| C37 | 0.018 | nd | | |
| C38 | 0.025 | nd | | |
| C39 | 0.031 | nd | | |
| C40 | 0.003 | nd | | |
| **PAHs** | 0.098 | 0.130 | 0.251 | 0.021 |
| Nap | 0.008 | 0.001 | 0.014 | 0.0004 |
| Acy | 0.005 | 0.0003 | 0.006 | 0.0002 |
| Ace | 0.001 | 0.00004 | 0.001 | 0.0003 |
| Flu | 0.002 | 0.0001 | | 0.001 |
| Phe | 0.005 | 0.021 | 0.007 | 0.008 |
| Ant | 0.001 | 0.001 | 0.002 | 0.0004 |



| | | | | | |
|---|---|---|---|---|---|
| Fluo | 0.002 | 0.009 | 0.027 | | 0.010 | 0.026 |
| Pyr | 0.007 | 0.008 | 0.052 | | 0.088 | 0.028 |
| BaA | 0.0005 | 0.001 | 0.014 | | 0.001 | 0.007 |
| Chry | 0.0005 | 0.003 | 0.025 | | 0.002 | 0.008 |
| BbF | 0.0003 | | 0.016 | | 0.001 | 0.002 |
| BkF | 0.0002 | | 0.003 | | 0.0001 | 0.001 |
| BaP | 0.0004 | | 0.009 | | 0.00001 | 0.0004 |
| IcdP | 0.001 | 0.0004 | 0.013 | | 0.00002 | 0.001 |
| DahA | 0.0002 | | 0.001 | | 0.001 | 0.000 |
| BghiP | 0.0003 | 0.0003 | 0.062 | | 0.004 | 0.003 |
| **Hopane, sterane** | | 0.143 | 0.167 | 0.014 | 0.011 | 0.026 |
| ABB | | | 0.007 | 0.0004 | 0.0005 | 0.001 |
| AAA | | | 0.006 | 0.001 | 0.001 | 0.002 |
| Tm | | 0.012 | 0.014 | 0.001 | 0.001 | 0.001 |
| 30AB | | 0.069 | 0.065 | 0.006 | 0.005 | 0.011 |
| 29AB | | 0.061 | 0.075 | 0.006 | 0.004 | 0.011 |





**Figure captions**

Figure 1 The routes for diesel trucks

Figure 2 Particulate matter sampling system

Figure 3 $EF_{PM}$ for excavators with different operating modes and emission standards

5     (a) and the correlation with sulfur contents (b)

Figure 4 Diesel trucks $EF_{PM}$ for different emission standards, vehicle sizes and driving

conditions

Figure 5 PM compositional constituents for individual vehicles (%)

Figure 6 OC/EC ratios under different operating modes and driving conditions for

10     excavators and trucks

Figure 7 Cross plots for the ratios of BaA/(BaA+Chry) vs IcdP/(IcdP+BghiP) and

BaA/(BaA+Chry) vs Flua/(Flua+Pry) and comparison with those from other diesel

vehicle sources.

Figure 8 BaPeq for parent PAHs in each tested excavators (A) and trucks (B)





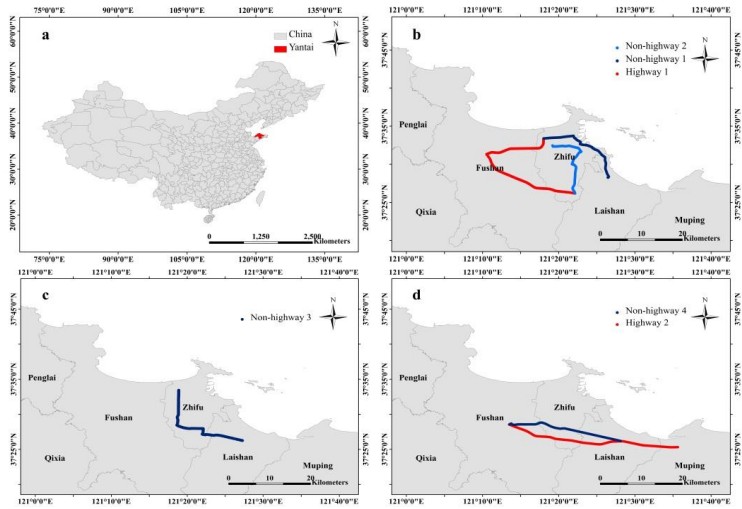

Figure 1The routes for diesel trucks; a was the site of Yantai; b was the route for
China Ⅲ and China Ⅳ light-duty diesel trucks; c was the rout for China Ⅱ
heavy-duty diesel truck; d was route for China Ⅲ medium-duty and heavy-duty trucks



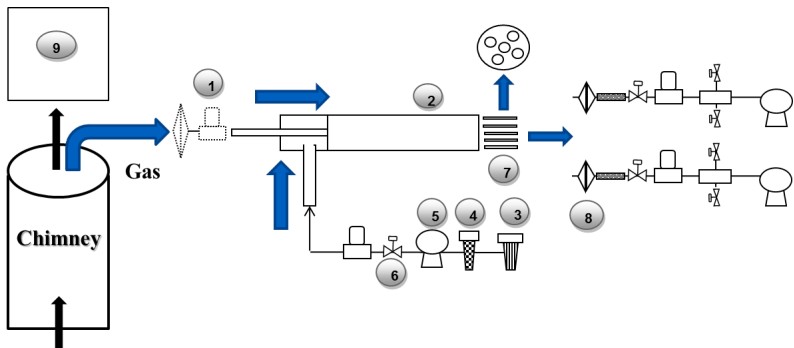

Figure 2 Particulate matter sampling system; 1 was flowmeter; 2 was dilute tunnel; 3 was filtrator; 4 was activated carbon; 5 was fan; 6 was valve; 7 was flow divider; 8 was filter membrane sampler; 9 was exhaust analyzer





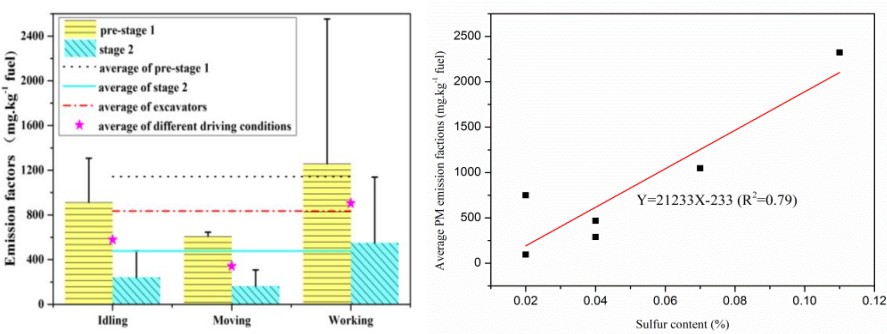

Figure 3 EF$_{PM}$ for excavators with different operating modes and emission standards

(a) and the correlation with sulfur contents (b)

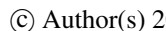



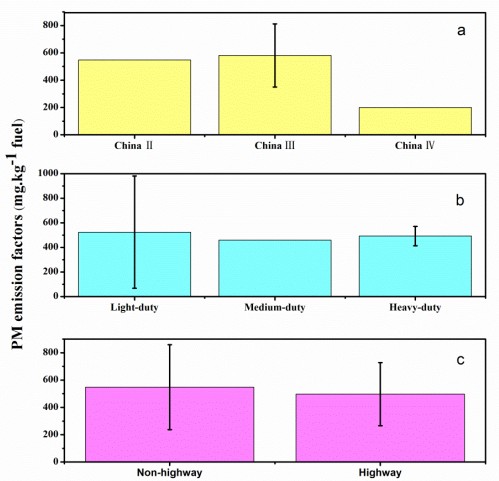

Figure 4 Diesel trucks EF$_{PM}$ for different emission standards (a), vehicle sizes (b) and driving conditions (c)





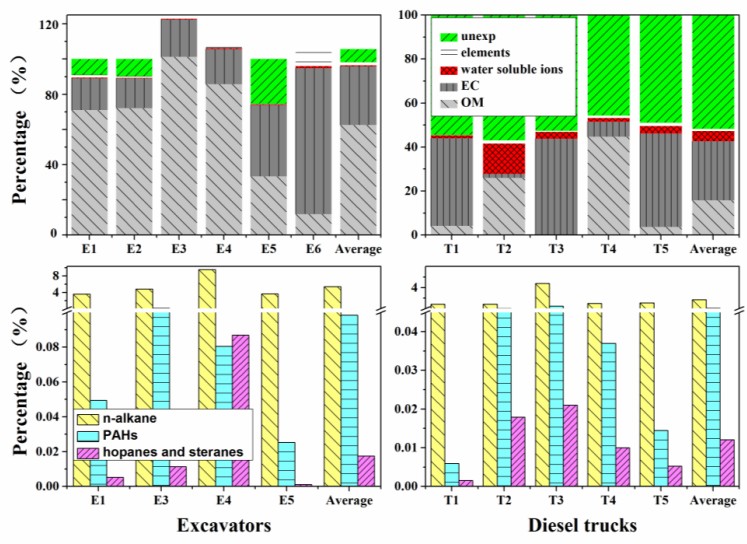

Figure 5 Compositional constituents of PM for individual vehicles (%)





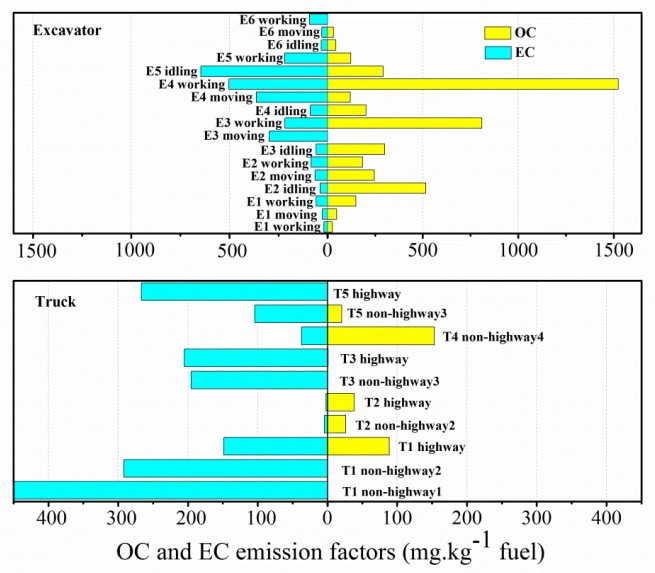

Figure 6 OC/EC ratios under different operating modes and driving conditions for

excavators and trucks





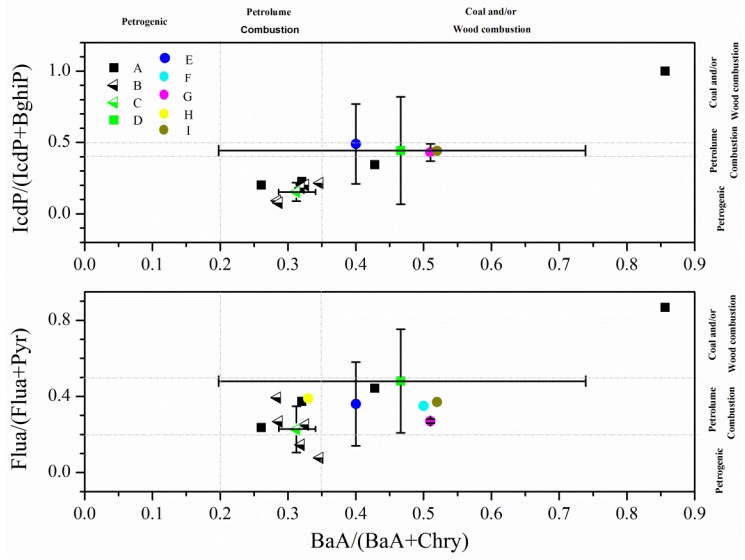

Figure 7 Cross plots for the ratios of BaA/(BaA+Chry) vs IcdP/(IcdP+BghiP) and BaA/(BaA+Chry) vs Flua/(Flua+Pry) and comparison with those from other diesel vehicle sources. A and B were isomer ratios of PAHs for excavators and trucks tested in this study, respectively; C and D were isomer ratios of PAHs for trucks and excavators tested in this study; E, F, G, H, I were results obtained from Liu et al (Liu et al., 2015), Wang et al (Wang et al., 2015), Shah et al (Shah et al., 2005), Schauer et al (Schauer et al., 1999), Chen et al (Chen et al., 2013)





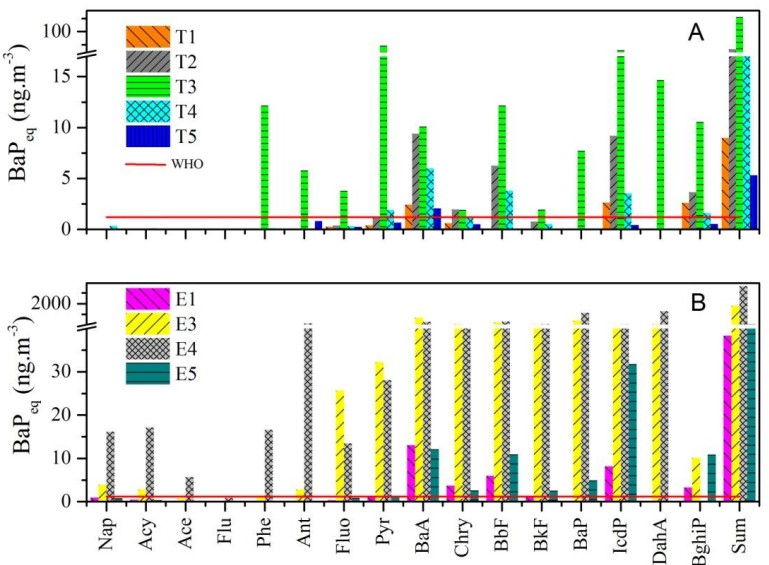

Figure 8 BaPeq for parent PAHs in each tested trucks (A) and excavators (B)