# Peer review of "Measurement of PM and its chemical composition in real-world emissions from non-road and on-road diesel vehicles"

_Atmospheric Chemistry and Physics, 2016_

## Referee Comment (RC1) · Anonymous Referee #1 · 25 Dec 2016

The manuscript by Cui et al. summarizes emissions measurements from multiple generation diesel excavators and trucks under different operating and driving conditions. These types of measurements are unique in China and much needed. The paper is well organized, but it needs a thorough edit as many words, verbs, etc are not used correctly or are missing. Below I highlight the technical weaknesses, minor clarifications, and instances where sentences are confusing and need to be rephrased. I approve publishing the paper after these concerns are addressed.

1. One of the weaknesses of this work is that each truck/excavator was tested only once. Thus it's unknown how representative these results are and how variability in the measurements affect the observed emission factors. I doubt that duplicate runs

can now be carried out; however, the authors should at least mention and address this weakness. Another weakness is that driving conditions of the trucks were not similar (as shown in Figure S2); since driving conditions and engine load can have significant impacts on the emission factors, how can the results be interpreted in a unified manner? This should also be addressed in the discussion and conclusion sections. Related to this is the variety of the engines tested in this work for both excavators and trucks. For example for excavators, engine powers span a range of 35-169 KW and total weights and engine displacements also vary a lot. On one hand, it's good to have a sampling pool of various engine types/sizes. On the other hand, these difference should be kept in mind and referred to when comparisons are made throughout the paper.

2. For readers who are not familiar with the standards in China, it will be useful to have a table where major particulate and gaseous emissions of each generation standard for trucks/excavators are listed.

3. P7, L23: Although mentioned in Table 2, please indicate in the text the average (or range of) sulfur content of the fuels as well as the limit of GB 252-2015.

4. P8, L 22: what recovery % for each species were achieved?

5. P12, L3: It seems the trucks with China II and China III standards had similar PM emission factors. Why is that so? Do these standards pose similar levels for PM? or is it that the trucks tested don't necessarily represent the standard? or is this an instance where results from a single measurement from a truck are uncertain?

6. P12, L7: unclear what "more volatile" means here

7. P12, L11-13: It doesn't make sense that trucks driven on road with higher grade have lower emissions. Please clarify.

8. P. 12, L17: what's the justification for using OM/OC=1.6 for such fresh emissions? How will the result change if a lower factor, more representative of fresh emissions, is

used?

9. P13, L9, P14,L2: it is mentioned that diesel sulfur content affected OC/EC. It is unclear to me how fuel sulfur can affect emission of organic compounds and soot. Please explain.

10. P13, L11-26: It is unclear what the elemental emissions are stemming from: the fuel or bad conditions of the engine or the lubricating oil? Please explain. For example, L22, it is mentioned that diesel quality used in E4 was poor. Was the fuel also tested for elemental content? Were Cu and Zn higher in this fuel as well?

11. P13, L23, P14, L4-5, P17, L24-26: Authors mention that % of elemental composition in E1 and E6 was higher. How did absolute concentrations or emission factors of the elements compared for these two vs. the others? Since % values depend on concentrations of other components as well, I don't think they're as relevant to be mentioned, especially since the contribute to a very small fraction of the emissions.

12. P14, L7-10: It is unclear how the authors concluded that alkane/hopane/steranes were influenced by fuel quality and PAHs by combustion. Please explain and clarify.

13. P16, L11: Please explain what reactions in the engine authors refer to.

14. P17, L3: Is it really that presence of metals oxidizes soot?! or do the metals enhance combustion and reduce formation of soot?

Minor comments: 1. Acronyms of PAHs should not be used in the abstract. 2. Define BaPeq in the abstract 3. P3, L 7: define PM. Throughout the paper indicate what size PM refers to (PM1, PM2.5, etc). 4. P12, L12: consider using "higher road grade" 5. P20, L3: Do authors mean excavators rather than diesel truck here or should E1, E2, .... be T1, T2, etc? 6. Figures: Axis labels are all too small and need to be modified for better quality figures. 7. Fig 7: what do the errors bars represent? Unclear form the caption what the difference between A-B and C-D symbols are. 8. Fig. S3. What are the crosses and dashed lines in these box and whisker plots?

Sentences needing to be rephrased: 1. P3, L 13-15 2. P4, L18-20 3. P7, L12-14 4. P12, L1-3 5. P. 13, L19-20 6. P.18, L3-4

---

## Referee Comment (RC2) · Anonymous Referee #2 · 3 Jan 2017

Cui et a. present data from measurements of particulate matter emissions and composition from real-world testing of a suite of on- and non-road diesel vehicles. They find that PM emissions, while variable, exhibit trends with fuel quality and emissions standard. Although these data add to the literature and will eventually help build more realistic emissions inventories for China, I do not recommend publication of this version of the manuscript in ACP. I have two major comments and numerous minor comments.

Major comments:

Fit: The manuscript, in my opinion, does not fit the research foci of publications typically accepted in ACP and I wonder if another journal would offer a better fit for this research. Here are my arguments against publication in ACP: (1) no new methods/instruments

were used that make the data novel, (2) the measurements were performed on a very small cross-section and are not necessarily representative of the on- and off-road fleet in China, (3) the small sample size, small cross-section, and large variability do not suggest large shifts/trends in emissions (or at least make them hard to observe), (4) comparisons with literature data are not very insightful. While the data add to the literature in terms of quantifying emission factors of PM from a modern set of vehicles under real-world conditions, the scientific contributions in this research effort are lean. The data need to be published but this journal may not be the right target.

Writing: The quality of technical communication is very poor. This suggests one or all of the following: (a) the first author was rushed to write and submit this manuscript, (b) the senior authors have not read through this manuscript, (c) the authors place no emphasis on clear and effective communication. The manuscript needs to be significantly improved by the senior authors to meet the expectations of an English language publication in a high impact journal. If the manuscript is not heavily edited for English, this would be reason enough for rejecting the manuscript from publication. Here are a few examples from just the first few pages: a. Page 1, line 24: 'involving wide-range emission standards'

b. Page 2, line 11: 'PM compositions emitted from excavators dominated'

c. Page 2, line 23: 'the complex of operating modes'

d. Page 3, line 7: 'diesel vehicles exhaust is a major source of emissions in ambient PM'

e. Page 3, line 9: '30% of emissions in ambient PM'

f. Page 3, line 18: 'causing severe emission situation'

g. Page 3, line 23: 'almost higher than 90% of PM came from on-road diesel vehicles emission'

h. Page 3, line 27: '349 thousand tons PM emission'
i. Page 5, line 23: 'organic matters'?

j. Page 5, line 26: 'impact factors of PM'; what does that mean?

Minor Comments:

1. Emissions standards: It might be worthwhile to describe the on-road and off-road emissions standards (e.g., Stages and China) and their emissions limits for PM (and other pollutants too) at the beginning of the manuscript through a Table. This would help orient the reader and also allow easy comparison with the EPA and EURO standards.

2. Page 2, line 9: Did vehicle exhaust contribute to 30% of the PM concentrations or emissions? Unclear; please clarify.

3. Page 4, line 3: construction equipment might be better word

4. Page 3, line 16 to page 4, line 5: It might be better if the number of vehicles, fuel consumption and PM emissions in China were represented through a table or figure, alongside the relative importance of trucks and excavators to justify the use of those vehicle types in this research.

5. Page 4, line 18 to page 5, 10: The authors have only cited other people's work but have not paraphrased their findings. Hence, it is unclear what the gaps and motivation for this work is.

6. Page 6, line 19: I did not understand how the duration of the different modes were determined. Also, what torque-speed ratings do the idling, moving, and working mode correspond to?

7. Page 7, line 28: Why did the researchers use quartz-fiber filters? My understanding is that the fibers can tear off during handling and bias the gravimetric measurement. Do the authors mean Teflon-coated quartz fiber filters?

8. Section 2.4.3: The BaPeq method needs to be discussed in detail for the reader to

follow the calculation.

9. Section 3.1: What fraction of the improvement between pre-stage 1 and stage 2 can be attributed to better quality fuel as opposed to the emission standard?

10. Section 3.2: Given that there was only one China IV truck, how confident are the authors in their assessment that China IV trucks are better compared to the China III trucks. Similarly, is the China II truck any different than the China III trucks. Can the authors comment on how the small sample size could affect their conclusion?

11. Section 3.3: Is the lack of a mass closure on the PM filter a result of using a quartz-fiber filter for gravimetric analysis?

12. Pry, Fluo etc.: Repeatedly, the authors have used abbreviated names to refer to various PM species. Using the full name of the species might improve readability.

13. Sections 3.3, 3.4 and 3.5: The authors have compared the PM composition data amongst the excavators and trucks and to literature data. However, it was hard for me to glean anything meaningful from all those comparisons and the ensuing discussion. I recommend that the authors spend some more time trying to make the interpretation more palatable to the reader.

14. Page 18, line 26 to page 19, line 2: The health relevant calculations, comparisons, and following discussion were too hard to follow and seemed like they were added to the manuscript as an afterthought.

---

## Author Comment (AC1) · 22 Feb 2017

Response to Referee' Comments on Manuscript: acp-2016-1038 Manuscript Number: acp-2016-1038 Title: Measurement of PM and its chemical composition in real-world emissions from non-road and on-road diesel vehicles Authors: Min Cui, Yingjun Chen, Yanli Feng, Cheng Li, Junyu Zheng, Chongguo Tian, Caiqing Yan, Mei Zheng Corresponding authors: Yingjun Chen, Yanli Feng, Junyu Zheng

Referee #1 General comments The manuscript by Cui et al. summarizes emissions measurements from multiple generation diesel excavators and trucks under different operating and driving conditions. These types of measurements are unique in China and much needed. The paper is well organized, but it needs a thorough edit as many

words, verbs, etc are not used correctly or are missing. Below I highlight the technical weaknesses, minor clarifications, and instances where sentences are confusing and need to be rephrased. I approve publishing the paper after these concerns are addressed.

Response: Thanks for the reviewer's positive approval. Clarifications have been provided and confusing sentences were rephrased in the revised manuscript.

Comments #1: (1) One of the weaknesses of this work is that each truck/excavator was tested only once. Thus it's unknown how representative these results are and how variability in the measurements affect the observed emission factors. I doubt that duplicate runs can now be carried out; however, the authors should at least mention and address this weakness. (2) Another weakness is that driving conditions of the trucks were not similar (as shown in Figure S2); since driving conditions and engine load can have significant impacts on the emission factors, how can the results be interpreted in a unified manner? This should also be addressed in the discussion and conclusion sections. (3) Related to this is the variety of the engines tested in this work for both excavators and trucks. For example for excavators, engine powers span a range of 35-169 KW and total weights and engine displacements also vary a lot. On one hand, it's good to have a sampling pool of various engine types/sizes. On the other hand, these difference should be kept in mind and referred to when comparisons are made throughout the paper.

Response: Thanks for the reviewer's constructive suggestions. This major question was divided into three questions and we would provide a personal response to your comments, separately.

(1)We appreciate the review's comment. Indeed, we are also attached importance to the weaknesses of tested time in this study. However, given the difficulties of field measurements and some important parameters missing in the links of repeat tests, only one relative complete test was chosen for further discussion. In order to evaluate

the variability, we had conducted some repeats for individual vehicles, and the results were presented in tables S3 and S4 in the revised supporting information. As shown in tables S3 and S4, the variability in test times for the same operational mode was considered acceptable. Moreover, we combined some repeat tests for organic matter analysis for T1 and T2, which could reduce the uncertainty. We confirmed that the weaknesses of repeatability existed in this study, and mentioned this weakness in the revised manuscript (Page 7 line 19-26).

(2) Thanks. As mentioned in the revised manuscript (Page 7 line 8-10), different emission standards diesel trucks must run on different roads, which was restricted by traffic rules. For example, "yellow label car" can only run on the particular road and is not allowed running on the highway and arterial road. Therefore, different routes were chosen for different trucks. Although driving conditions of the trucks were not similar shown in Figure S2, the different characteristics of velocity on the highway and non-highway were obviously. Therefore, we just discussed highway and non-highway routes in this study. We have addressed this weakness and interpreted the unified manner in the revised manuscript(Page 7 line 11-12).

(3) Thanks for the comment. As we could seen from Figure S5 in the revised supporting information, the average EFPM was less affected by engine power. It was regretful that the sample size in this study seemed not enough to reflect the impact from engine power. Thus, we just gave EFPM of different engine power in the revised manuscript, and didn't discuss in-depth (Page 11 line 27-29). Comments #2:For readers who are not familiar with the standards in China, it will be useful to have a table where major particulate and gaseous emissions of each generation standard for trucks/excavators are listed.

Response: Thanks for the suggestion. The major particulate and gaseous emissions of each generation standard for trucks/excavators were listed in Tables S1 and S2 (Supporting information).

Comments #3: P7, L23: Although mentioned in Table 2, please indicate in the text the average (or range of) sulfur content of the fuels as well as the limit of GB 252-2015.

Response: Thanks. The range of sulfur content and limit of GB 252-2015 have been added in the revised manuscript (Page 8 line 19-20).

Comments #4: P8, L22: what recovery % for each species were achieved?

Response: Thanks for the comment. The recoveries of five surrogates have been added in the revised manuscript (Page 9 line 21).

Comments #5:P12, L3: It seems the trucks with China II and China III standards had similar PM emission factors. Why is that so? Do these standards pose similar levels for PM? or is it that the trucks tested don't necessarily represent the standard? or is this an instance where results from a single measurement from a truck are uncertain?

Response: We appreciate the review's comment. As we discussed in the manuscript, the most important reason causing this result was different driving conditions for those two trucks. Due to heavy pollutions from China II trucks, traffic laws regulate that China II trucks are forbidden to drive on city center and only allowed to drive on some remote parts of the city, while the roads for China III trucks are always jammed. For evaluating the emission from trucks in the real world, we shouldn't neglect the driving conditions to discuss trucks itself. However, we confirmed that the number of measurement was shortage in this study, and we will lucubrate in the future.

Comments #6:P12, L7: unclear what "more volatile" means here

Response: Thanks for the comment. "more volatile" refers to highly varied speed (Page 13 line 11).

Comments #7:P12, L11-13: It doesn't make sense that trucks driven on road with higher grade have lower emissions. Please clarify.

Response: Thanks. There was wrong with expression and we have modified in the

revised manuscript (Page 13 line 16).

Comments #8:P. 12, L17: what's the justification for using OM/OC=1.6 for such fresh emissions? How will the result change if a lower factor, more representative of fresh emissions, is used?

Response: Thanks for the reviewer's constructive suggestions. Chow et al (2015) showed that a conversion factor used to transform OC to OM was ranged from 1.2 to 2.6, depending on the extend of OM oxidation. Fresh aerosols from different sources had different values, such as 1.4 and 1.6 for diesel engine (Gilardoni et al., 2007, Japar et al., 1984) and 1.7 for biomass burning (Chow et al., 2015). Therefore, we assumed the conversion factor is 1.6 in this study.

Comments #9:P13, L9, P14,L2: it is mentioned that diesel sulfur content affected OC/EC. It is unclear to me how fuel sulfur can affect emission of organic compounds and soot. Please explain.

Response: We appreciate the review's comment. According to references, we assumed that the formation of organic compounds and soot was obviously affected by diesel sulfur content in two points. On the one hand, organosulfurs constituted up to 62% of the total sulfur content in diesel (Adlakha et al., 2016). Organic compounds existing in diesel were removed simultaneously by process of desulfurization. Therefore, emissions of organic compounds and soot generated by hydrogen abstraction/acetylene addition were reduced (Sánchez et al., 2013). One the other hand, sulfuric acid, the nucleating agent in diesel particle formation, generated by sulfur in diesel (Ruiz et al., 2015). These nucleating agents might provide a place for organic compounds condensation and reaction.

Comments #10:P13, L11-26: It is unclear what the elemental emissions are stemming from: the fuel or bad conditions of the engine or the lubricating oil? Please explain. For example, L22, it is mentioned that diesel quality used in E4 was poor. Was the fuel also tested for elemental content? Were Cu and Zn higher in this fuel as well?

[ACPD Interactive comment / Printer-friendly version / Discussion paper]

[Figure]

Response: Thanks for the comment. Although diesel quality was analyzed in this study, many elemental contents were below the method detection limit. Wang et al. (2003) reported that the concentrations of Fe, Ca and Mg accounted for 50% of the total elements in diesel fuel. Thus, the possible source of elements was diesel, while Cu and Zn were affected by sampling environments for E4. The detail information could be seen in the revised manuscript (Page 14 line 23-30).

Comments#11:P13, L23, P14, L4-5, P17, L24-26: Authors mention that % of elemental composition in E1 and E6 was higher. How did absolute concentrations or emission factors of the elements compared for these two vs. the others? Since % values depend on concentrations of other components as well, I don't think they're as relevant to be mentioned, especially since the contribute to a very small fraction of the emissions.

Response: Thanks for the comment. The average emission factors of elemental were 5.66 mgÂůkg-1 for E1+E6 and 4.02 mgÂůkg-1 for E2+E3+E4+E5, and were mentioned in the revised manuscript (Page 15 line 2-3).

Comments#12:P14, L7-10: It is unclear how the authors concluded that alkane/hopane/steranes were influenced by fuel quality and PAHs by combustion. Please explain and clarify.

Response: Thanks for the comment. N-alkanes, hopanes and steranes fractions were the highest in excavator E4, while PAHs fraction was the highest in excavator E3. Comparing the fuel quality between E3 and E4, E4 had a poorer diesel quality, which might be the main reason for high n-alkane, hopanes and steranes. Similarly, it was said by Rogge et al. (1993) that n-alkanes, hopanes and steranes were mostly derived from incomplete combustion of fuel and lubricant oil. However, we speculated that PAHs was affected by combustion conditions (e.g. combustion temperature) in this study, due to E3's better performance (stage 2) and relatively superior fuel quality. The distinct explanation was added in the revised manuscript (Page 15 line 12-21).

Comments#13:P16, L11: Please explain what reactions in the engine authors refer to.

Response: Thanks for the comment. The description of reactions was provided in the revised manuscript (Page 17 line 19-20).

Comments#14:P17, L3: Is it really that presence of metals oxidizes soot?! or do the metals enhance combustion and reduce formation of soot?

Response: We appreciate the review's comment. It was said by Kasper et al., (1999) that the action of iron oxide was recognized as a catalyst and burnout rate of soot could promote during combustion process. Therefore, we inferred that metals may enhance combustion of soot. The corresponding expression was added in the revised manuscript (Page 18 line 10-12).

Comments#15: Acronyms of PAHs should not be used in the abstract.

Response: Thanks, the acronyms of PAHs have been changed to full names (Page 1 line 24; Page 2 line 25-27).

Comments#16:Define BaPeq in the abstract 3.

Response: Thanks for the comment. The BaPeq has been defined in the revised abstract (Page 3 line 1).

Comments#17: P3, L 7: define PM. Throughout the paper indicate what size PM refers to (PM1, PM2.5, etc).

Response: Thanks for the comment. PM referred to total suspended particulate (Dp$\leq$100 $\mu$m) in this study. We have remarked in the revised manuscript (Page 3 line 12).

Comments#18: P12, L12: consider using "higher road grade".

Response: Thanks. We agree with the reviewer's suggestion and changed the word as suggested (Page 13 line 16).

Comments#19: P20, L3: Do authors mean excavators rather than diesel truck here or

should E1, E2,.... be T1, T2, etc?

Response: Thanks. The E1,E2... have changed to T1,T2....(Page 21 line 15-16)

Comments#20: Figures: Axis labels are all too small and need to be modified for better quality figures.

Response: Thanks for the advice. Axis labels in Figure 1, 3, 4, 5 and 7 were modified in the revised manuscript (Page 35; Page 37; Page 38; Page 39; Page 41).

Comments#21:Fig 7: what do the errors bars represent? Unclear form the caption what the difference between A-B and C-D symbols are.

Response: Thanks for the comment. A and B are isomer ratios of PAHs for excavators and trucks tested in this study, respectively; C and D are average isomer ratios of PAHs for trucks and excavators tested in this study. The vertical and horizontal errors bars represent the standard deviation of values shown in vertical and horizontal axis, respectively (Page 41).

Comments#22:Fig. S3. What are the crosses and dashed lines in these box and whisker plots?

Response: Thanks for the comment. The annotations wre shown in the revised Figure S4 (Supporting information).

Comments#23:Sentences needing to be rephrased: 1. P3, L 13-15 2. P4, L18-20 3. P7, L12-14 4. P12, L1-3 5. P. 13, L19-20 6. P.18, L3-4.

Response: Thanks for the comment. We have made every effort to polish our English and asked a native English speaker to take a proof reading of the revised manuscript.

References: Adlakha, J., Singh, P., Ram, S.K., Kumar, M., Singh, M.P., Singh, D., Sahai, V., Srivastava, P.: Optimization of conditions for deep desulfurization of heavy crude oil and hydrodesulfurized diesel by Gordonia sp. IITR100, Fuel. 184: 761-769 2016 Chow, J.C., Lowenthal, D.H., Chen, L.W.A., Wang, X.L., Watson, J.G.: Mass

reconstruction methods for PM2.5: a review, Air Quality Atmosphere and Health. 8(3): 243-263 2015 Gilardoni, S., Russell, L.M., Sorooshian, A., Flagan, R.C., Seinfeld, J.H., Bates, T.S., Quinn, P.K., Allan, J.D., Williams, B., Goldstein, A.H., Onasch, T.B., Worsnop, D.R.: Regional variation of organic functional groups in aerosol particles on four US east coast platforms during the International Consortium for Atmospheric Research on Transport and Transformation 2004 campaign, Journal of Geophysical Research-Atmospheres. 112(D10): 11 2007 Japar, S.M., Szkarlat, A.C., Gorse, R.A., Heyerdahl, E.K., Johnson, R.L., Rau, J.A., Huntzicker, J.J.: comparison of solvent-extraction and thermal optical carbon analysis-methods - application to diesel vehicle exhaust aerosol, Environmental Science & Technology. 18(4): 231-234 1984 Kasper, M., Sattler, K., Siegmann, K., Matter, U., Siegmann, H.C.: The influence of fuel additives on the formation of carbon during combustion, Journal of Aerosol Science. 30(2): 217-225 1999 Rogge, W.F., Hildemann, L.M., Mazurek, M.A., Cass, G.R., Simoneit, B.R.T.: Sources of fine organic aerosol. 2. Noncatalyst and catalyst-equipped automobiles and heavy-duty diesel trucks, Environmental Science & Technology. 27(4): 636-651 1993 Ruiz, F.A., Cadrazco, M., López, A.F., Sanchez-Valdepeñas, J., Agudelo, J.R.: Impact of dual-fuel combustion with n-butanol or hydrous ethanol on the oxidation reactivity and nanostructure of diesel particulate matter, Fuel. 161: 18-25 2015 Sánchez, N.E., Millera, Á., Bilbao, R., Alzueta, M.U.: Polycyclic aromatic hydrocarbons (PAH), soot and light gases formed in the pyrolysis of acetylene at different temperatures: Effect of fuel concentration, Journal of Analytical and Applied Pyrolysis. 103: 126-133 2013 Wang, Y.F., Huang, K.L., Li, C.T., Mi, H.H., Luo, J.H., Tsai, P.J.: Emissions of fuel metals content from a diesel vehicle engine, Atmospheric Environment. 37(33): 4637-4643 2003

Please also note the supplement to this comment:
http://www.atmos-chem-phys-discuss.net/acp-2016-1038/acp-2016-1038-AC1-supplement.zip

Table S3 Pollutants mass concentrations emitted from E4 in three idling repeat tests

| | $O_2{}^a$ (%) | $CO_2{}^a$ (%) | $CO^a$ (ppm) | $NOx^a$ (ppm) | PM (mg m$^{-3}$) | OC (mg m$^{-3}$) | EC (mg m$^{-3}$) |
|---|---|---|---|---|---|---|---|
| 1 | 16.2 | 3.4 | 309 | 453 | 11.9 | 4.3 | 1.9 |
| 2 | 16.3 | 3.4 | 257 | 457 | 14.6 | 6.1 | 2.9 |
| 3 | 16.3 | 3.4 | 262 | 445 | 14.4 | 6.8 | 2.5 |
| SD | 0.08 | 0.01 | 28.6 | 5.68 | 1.55 | 1.26 | 0.53 |

a: the datum were presented on other unpublished research

**Fig. 1.** Table S3 Pollutants mass concentrations emitted from E4 in three idling repeat tests

Table S4 PM mass concentrations emitted from trucks in some repeat tests (mg m$^{-3}$)

| Trucks | Roads | 1 | 2 | 3 | SD |
|---|---|---|---|---|---|
| Light duty-China III | non-highway 1 | 15.0 | 16.2 | / | 0.87 |
| | highway 1 | 19.8 | 30.6 | / | 7.67 |
| | non-highway 2 | 21.3 | 16.1 | / | 3.68 |
| Heavy duty-China II | non-highway 3 | 7.87 | 6.11 | 6.69 | 0.89 |
| Medium duty-China III | non-highway 4 | 11.0 | 10.3 | / | 0.49 |
| | highway 2 | 8.79 | 17.1 | / | 5.85 |
| Heavy duty-China III | non-highway 4 | 5.29 | 9.56 | 6.99 | 2.15 |
| | highway 2 | 10.6 | 7.42 | / | 2.24 |

**Fig. 2.** Table S4 PM mass concentrations emitted from trucks in some repeat tests (mgÂům-3)

[Figure]

Figure S5 PM emission factors for different power excavators

**Fig. 3.** Figure S5 PM emission factors for different power excavators

---

## Author Comment (AC2) · 22 Feb 2017

Response to Referee' Comments on Manuscript: acp-2016-1038 Manuscript Number: acp-2016-1038 Title: Measurement of PM and its chemical composition in real-world emissions from non-road and on-road diesel vehicles Authors: Min Cui, Yingjun Chen, Yanli Feng, Cheng Li, Junyu Zheng, Chongguo Tian, Caiqing Yan, Mei Zheng Corresponding authors: Yingjun Chen, Yanli Feng, Junyu Zheng

Referee #2: General comments Cui et al. present data from measurements of particulate matter emissions and composition from real-world testing of a suite of on- and non-road diesel vehicles. They find that PM emissions, while variable, exhibit trends with fuel quality and emissions standard. Although these data add to the literature and

will eventually help build more realistic emissions inventories for China, I do not recommend publication of this version of the manuscript in ACP. I have two major comments and numerous minor comments.

Response: Thanks very much for the comments. We have revised this manuscript carefully, and please find our detailed responses below.

Major comments: Fit: The manuscript, in my opinion, does not fit the research foci of publications typically accepted in ACP and I wonder if another journal would offer a better fit for this research.

Response: Thanks. But the authors disagree with this comment and consider ACP as the best journal of high quality to publish our precise measurement data. On the one hand, in recent years, PM emission from diesel vehicles drawn more and more attention in China, due to severe air pollution. However, the great uncertainty existing in PM from diesel vehicles exhausts makes those field datum very precious. Although our research is preliminary, as far as we know, this manuscript is the first on-board research in China that focused on PM chemical constituents from on-road and non-road diesel vehicles exhaust. The results of this study could provide basic data for air quality assessment and establishment of emission standard. Therefore, we chose ACP, one of the most influential journals in atmospheric fields, to publish our results for obtaining broader attention. On the other hand, the main subject areas for ACP comprise atmosphere modeling, field measurements, remote sensing, and laboratory studies of gases, aerosols, etc. Nowadays, several researches about emission factors and characteristics of PM from diesel engine have been published in ACP (Dai et al., 2015, Dallmann et al., 2014, Lin et al., 2015, Zhang et al., 2015). Therefore, this manuscript is fit to publish in ACP, because of general implications for source apportionment and health assessment.

Comments #1:No new methods/instruments were used that make the data novel.

Response: Thanks. We have added some descriptions about the progressiveness of

methods/instruments used which made the data novel in the present study in the revised manuscript (Page 8 line 26-27; Page 9 line 4-9). Briefly, the portable on-board emission measurement and dilute sampling system which was designed and manufactured in our laboratory has good performance (Zhang et al., 2015), and has obvious advancement compared with other on-board instrument for vehicles such as PEMS and FPS4000 (Zheng et al., 2015) by the portability and capability of filter sample collection for further PM chemical analysis in the laboratory. Furthermore, the present result was the first set data of on-board measurement for non-road diesel vehicle exhaust in China.

Comments #2:The measurements were performed on a very small cross-section and are not necessarily representative of the on- and off-road fleet in China. The small sample size, small cross-section, and large variability do not suggest large shifts/trends in emissions (or at least make them hard to observe).

Response: Thanks. We admitted that the sample size in this study was small, but wide ranges of vehicle types (including different emission standards and engine powers) were considered in this study. Furthermore, the most important purpose in this manuscript was to analyze the chemical constituents of PM from diesel vehicles exhausts, which needed a heavy workload (Page 7 line 3-5). Actually, we had selectively conducted some repeated experiments in this study to evaluate those variability and the results were shown in Tables S3 and S4 (Supporting information). As shown in the Tables S3 and S4, the variability was considered acceptable. Because there were some parameters missing in the field measurement, we decided to select an completed test for calculating the emission factors and combine the repeated filters to reduce this uncertain for some diesel vehicles. In the future work, we would increase the sample size to ensure the datum stability after this first attempt (Page 7 line 18-25).

Comments #3:Comparisons with literature data are not very insightful. While the data add to the literature in terms of quantifying emission factors of PM from a modern set of vehicles under real-world conditions, the scientific contributions in this research effort

are lean. The data need to be published but this journal may not be the right target.

Response: Thanks. The purpose and the greatest contribution of this study were established characteristics of PM and its constituents emitted from trucks and excavators using on-board measurements. In China, diesel vehicles are facing imperfect emission standards and messy diesel quality, especially for non-road diesel vehicles. The knowledge relative to the characteristics of PM emission from those diesel vehicles was slim to none. It was extremely difficult to collect literature data and compare with results obtained in this study, due to lacking of researches for characteristics of PM and its constituents by on-board tests. Following the reviewer's suggestion, we have made more interpretation among the comparison in the revised manuscript (Page 14 line 23-30; Page 15 line 12-21; Page 19 line 30; Page20 line 1-9). Finally, we chose ACP to publish our results for obtained broader attention from the perspective of the importance of the datum.

Comments #4:Writing: The quality of technical communication is very poor. This suggests one or all of the following: (a) the first author was rushed to write and submit this manuscript, (b) the senior authors have not read through this manuscript, (c) the authors place no emphasis on clear and effective communication. The manuscript needs to be significantly improved by the senior authors to meet the expectations of an English language publication in a high impact journal. If the manuscript is not heavily edited for English, this would be reason enough for rejecting the manuscript from publication. Here are a few examples from just the first few pages: a. Page 1, line 24: 'involving wide-range emission standards' b. Page 2, line 11: 'PM compositions emitted from excavators dominated' c. Page 2, line 23: 'the complex of operating modes' d. Page 3, line 7: 'diesel vehicles exhaust is a major source of emissions in ambient PM' e. Page 3, line 9: '30% of emissions in ambient PM' f. Page 3, line 18: 'causing severe emission situation' g. Page 3, line 23: 'almost higher than 90% of PM came from on-road diesel vehicles emission' h. Page 3, line 27: '349 thousand tons PM emission' i. Page 5, line 23: 'organic matters'? j. Page 5, line 26: 'impact factors of PM'; what does

that mean?

Response: Thanks for the comment. We have made every effort to polish our English and asked a native English speaker to take a proof reading of the final version of the revised manuscript. a 'Involving wide-range emission standards' was changed to 'involving a range of emission standards. b 'PM compositions emitted from excavators dominated' was changed to 'PM composition emitted from excavators was dominated'. c 'The complex of operating modes' was changed to 'the complex characteristics of excavator operational modes'. d 'Diesel vehicles exhaust is a major source of emissions in ambient PM' was changed to 'Diesel vehicles exhaust is a major source of ambient PM emissions'. e '30% of emissions in ambient PM' was changed to '30% of ambient PM emissions'. f 'Causing severe emission situation' was changed to 'and have contributed to severe emissions problems'. g 'Almost higher than 90% of PM came from on-road diesel vehicles emission' was changed to 'more than 90% of PM resulted from on-road diesel vehicle emissions'. h '349 thousand tons PM emission' was changed to '349 Gg of PM emissions'. i 'Organic matters' was changed to 'organic compounds'. j 'Impact factors of PM' was changed to 'influential factors of PM'.

Minor Comments:

Comments #1: Emissions standards: It might be worthwhile to describe the on-road and off-road emissions standards (e.g., Stages and China) and their emissions limits for PM (and other pollutants too) at the beginning of the manuscript through a Table. This would help orient the reader and also allow easy comparison with the EPA and EURO standards.

Response: Thanks. We have added the on-road and off-road emission standards in the revised manuscript (Supporting information).

Comments #2: Page 2, line 9: Did vehicle exhaust contribute to 30% of the PM concentrations or emissions? Unclear; please clarify.

Response: Thanks. We have modified the unclear place in the revised manuscript (Page 3 line 9-10).

Comments #3: Page 4, line 3: construction equipment might be better word

Response: Thanks for the advice. Following the reviewer's suggestion, we have changed the expression in revised manuscript (Page 4 line 12).

Comments #4: Page 3, line 16 to page 4, line 5: It might be better if the number of vehicles, fuel consumption and PM emissions in China were represented through a table or figure, alongside the relative importance of trucks and excavators to justify the use of those vehicle types in this research.

Response: Thanks for the advice. The figure S1 was added in the revised supporting information (Supporting information).

Comments #5: Page 4, line 18 to page 5, 10: The authors have only cited other people's work but have not paraphrased their findings. Hence, it is unclear what the gaps and motivation for this work is.

Response: Thanks. We have rephrased the correspond contents in revised manuscript (Page 5 line 7-25).

Comments #6: Page 6, line 19: I did not understand how the duration of the different modes were determined. Also, what torque-speed ratings do the idling, moving, and working mode correspond to?

Response: Thanks. The time of sampling under different modes was not strictly required, as long as assured enough contents of PM to conduct chemical analysis. We have clarified it in the revised manuscript (Page 7 line 1-5). Actually, the basis of selecting those modes were not according to torque-speed ratings. The idling mode refers to engine keeps running at low speed (about 600-800 rpm), but not moving or working. The moving mode refers to that excavator moves at low speed (below 3-5 kmÂůh-1) , but the bucket is not unload. The working mode refers to that bucket scoops the soil,

then moves to another location and scoops again.

Comments #7: Page 7, line 28: Why did the researchers use quartz-fiber filters? My understanding is that the fibers can tear off during handling and bias the gravimetric measurement. Do the authors mean Teflon-coated quartz fiber filters?

Response: Thanks. We used quartz-fiber filters for gravimetric measurements in this study. The quartz-fiber filters losses could be neglected. Because the filters were parceled by aluminum foil after sampling to avoid filters tearing off, and the PM weight of error in quartz-fiber and Teflon filters could acceptance. In adition, quartz-fiber filters were selected to measure PM weight for consistent with those used in the chemical analysis. We have added the reasons in the revised manuscript (Page 8 line 25-28).

Comments #8: Section 2.4.3: The BaPeq method needs to be discussed in detail for the reader to follow the calculation.

Response: Thanks. The detailed BaPeq method was added in the revised manuscript (Page 11 line 8-16).

Comments #9: Section 3.1: What fraction of the improvement between pre-stage 1 and stage 2 can be attributed to better quality fuel as opposed to the emission standard?

Response: Thanks. We supposed that the fuel quality rather than the emission standards has a more great impact on PM constituents. Although the threshold (total emission) was set in non-road emission standards, constitutes of PM haven't regulated in these standards. Furthermore, it was said that sulfur in fuel translates to sulfuric acid which is the nucleating agent in diesel nanoparticle formation (Ruiz et al., 2015). After sulfuric acid nucleation particles formation, the organic compounds (volatile and low volatile) condense on it. Similarity, the soot was also influenced by this nucleating agent (Schneider et al., 2005). Considering the limit of sample size of our study, it was difficult to calculate the influence of the fuel quality and the emission standards on PM constituents separately. In our future study, we will continue to focus on this complex

issue.

Comments #10: Section 3.2: Given that there was only one China IV truck, how confident are the authors in their assessment that China IV trucks are better compared to the China III trucks. Similarly, is the China II truck any different than the China III trucks. Can the authors comment on how the small sample size could affect their conclusion?

Response: We appreciated this question. Actually, China IV truck is extremely rare, because few trucks could reach this emission standards in China. Therefore, we just found only one truck of China IV to conduct experimental. Furthermore, through comparing our results with references and assessing repeatability in the test results, we considered that our conclusions were credible. The detail explanations were added in the revised manuscript (Page 7 line 22-26; Page 16 line 20-21).

Comments #11: Section 3.3: Is the lack of a mass closure on the PM filter a result of using a quartz-fiber filter for gravimetric analysis?

Response: Thanks. We have replied in the comment 7, using quartz-fiber filter was not the main reason caused poor mass closure. The main reasons might be distribution error from OC and EC, water effect and metal oxidation. As mentioned in the revised manuscript, the distribution error from OC and EC by using IMPROVE could highly affect the results of mass closure (Page 16 line 17-19). As shown in Table R1, emission factors of OC was lower than those of n-alkanes for T3, which indicated that the OC content was underestimated. For example, emission factors of OC increased to 85.0 mgÂůkg-1 fuel, the mass closure would almost increase by 10%, correspondingly. For T2, the thick moisture was trapped in the filter, which could increase PM weighing error.

Comments #12: Pry, Fluo etc.: Repeatedly, the authors have used abbreviated names to refer to various PM species. Using the full name of the species might improve readability.

Response: Thanks for the comment. The full name of the individual PAH was displayed

in the revised manuscript (Page 9 line 26-30). But, considering the concise expression, we also used abbreviated names in the part of discussion.

Comments #13: Sections 3.3, 3.4 and 3.5: The authors have compared the PM composition data amongst the excavators and trucks and to literature data. However, it was hard for me to glean anything meaningful from all those comparisons and the ensuing discussion. I recommend that the authors spend some more time trying to make the interpretation more palatable to the reader.

Response: We appreciate the review's comment. We also want to do it, but the maneuverability was poor. It is extremely difficult to collect literature data and compare with results obtained in this study, due to lacking of researches for characteristics of PM and its constituents by on-board tests, especially for non-road diesel engine. Based on our purpose in this manuscript, we presented three parts for further discussion. In section 3.3, we tried to interpret difference in characteristics of PM emission between individual diesel vehicles tested in this study. In section 3.4, we tried to combine our results with those from other references to find some consensus. In section 3.5, through comparing the differences in characteristic of PM emission between excavators and trucks, we emphasized the PM emission difference of two types of vehicles. Following the reviewer's suggestion, we have made more interpretation between the comparison in the revised manuscript (Page 14 line 23-30; Page 15 line 12-21; Page 19 line 30; Page 20 line 1-9).

Comments #14: Page 18, line 26 to page 19, line 2: The health relevant calculations, comparisons, and following discussion were too hard to follow and seemed like they were added to the manuscript as an afterthought.

Response: Thanks. The carcinogenic risks of PAHs emitted from trucks and excavators were the important indicators to evaluate emission situation for those two diesel vehicles. We have enhanced the expression in the revised manuscript (Page 11 line 8-16; Page 19 line 30; Page20 line 1-9).

References: Dai, S., Bi, X., Chan, L.Y., He, J., Wang, B., Wang, X., Peng, P., Sheng, G., Fu, J.: Chemical and stable carbon isotopic composition of PM2.5 from on-road vehicle emissions in the PRD region and implications for vehicle emission control policy, Atmospheric Chemistry and Physics. 15(6): 3097-3108 2015 Dallmann, T.R., Onasch, T.B., Kirchstetter, T.W., Worton, D.R., Fortner, E.C., Herndon, S.C., Wood, E.C., Franklin, J.P., Worsnop, D.R., Goldstein, A.H., Harley, R.A.: Characterization of particulate matter emissions from on-road gasoline and diesel vehicles using a soot particle aerosol mass spectrometer, Atmospheric Chemistry and Physics. 14(14): 7585-7599 2014 Lin, Y.C., Tsai, C.J., Wu, Y.C., Zhang, R., Chi, K.H., Huang, Y.T., Lin, S.H., Hsu, S.C.: Characteristics of trace metals in traffic-derived particles in Hsuehshan Tunnel, Taiwan: size distribution, potential source, and fingerprinting metal ratio, Atmospheric Chemistry and Physics. 15(8): 4117-4130 2015 Ruiz, F.A., Cadrazco, M., López, A.F., Sanchez-Valdepeñas, J., Agudelo, J.R.: Impact of dual-fuel combustion with n-butanol or hydrous ethanol on the oxidation reactivity and nanostructure of diesel particulate matter, Fuel. 161: 18-25 2015 Schneider, J., Hock, N., Weimer, S., Borrmann, S., Kirchner, U., Vogt, R., Scheer, V.: Nucleation Particles in Diesel Exhaust: Composition Inferred from In Situ Mass Spectrometric Analysis, Environmental Science & Technology. 39(16): 6153-6161 2005 Zhang, F., Chen, Y., Tian, C., Li, J., Zhang, G., Matthias, V.: Emissions factors for gaseous and particulate pollutants from offshore diesel engine vessels in China, Atmos. Chem. Phys. Discuss., 15(17): 23507-23541 2015 Zheng, X., Wu, Y., Jiang, J.K., Zhang, S.J., Liu, H., Song, S.J., Li, Z.H., Fan, X.X., Fu, L.X., Hao, J.M.: Characteristics of On-road Diesel Vehicles: Black Carbon Emissions in Chinese Cities Based on Portable Emissions Measurement, Environmental Science & Technology. 49(22): 13492-13500 2015

Please also note the supplement to this comment:
http://www.atmos-chem-phys-discuss.net/acp-2016-1038/acp-2016-1038-AC2-supplement.zip

[Figure]

Table R1 Mass closure on the PM filter for trucks

| Species | Units | T1 | T2 | T3 | T4 | T5 |
|---|---|---|---|---|---|---|
| OC | mg/kg fuel | 22.4 | 32.7 | 0.64 | 153 | 10.3 |
| EC | mg/kg fuel | 337 | 3.61 | 200 | 37.4 | 186 |
| OM | mg/kg fuel | 35.9 | 52.3 | 1.02 | 245 | 16.5 |
| Water soluble ions | mg/kg fuel | 12.0 | 27.7 | 14.5 | 8.80 | 14.6 |
| Elements | mg/kg fuel | 0.77 | 2.95 | 2.15 | 6.34 | 6.62 |
| N-alkanes | mg/kg fuel | 7.19 | 1.79 | 4.72 | 26.2 | 4.87 |
| PAHs | mg/kg fuel | 0.05 | 0.11 | 0.17 | 2.94 | 0.06 |
| Hopane and sterane | mg/kg fuel | 0.01 | 0.03 | 0.05 | 0.12 | 0.02 |
| PM | mg/kg fuel | 847 | 200 | 459 | 548 | 436 |
| Mass balance | % | 46 | 43 | 49 | 54 | 53 |

**Fig. 1.** Table R1 Mass closure on the PM filter for trucks

---

## Author Response (AR2)

**Response to Reviewers' Comments on Manuscript: acp-2016-1038**

Dear Gregory Frost,

Thank you so much for your consideration! Also, the anonymous reviewer's comments are highly appreciated! So far, we have revised the manuscript accordingly. Our point-by-point responses (in blue) to each reviewer's comments are listed below. And the modifications in the revised manuscript with marks are marked in red. Please see the manuscript for details.

Thanks a ton!

Sincerely,

Dr. Yingjun Chen

**Referee #1**

General comments

The authors have addressed most of my concerns in this revised manuscript. Quality of the presentation and text is much improved, although as mentioned below some proofreading is still needed. I still have other concerns about the dataset and presentation of the results that are highlighted below; therefore, I cannot support publishing the manuscript as is.

My most important comment is related to the sample size and conclusions drawn from the results (#1 below).

Comments #1: In response to comment #9 from the 2nd referee, the authors mentioned that "Considering the limit of sample size of our study, it was difficult to calculate the influence of the fuel quality and the emission standards on PM constituents separately."; Yet, in abstract (and p. 12, line 12-13) it is suggested that the lower $EF_{PM}$ in stage 2 compared to pre-stage 1 is due to the regulations. A few

sentenced lower, it's also mentioned that "Although the $EF_{PM}$ for excavators and trucks was reduced by the constraint of stringent emission standards,...". P11, line 23-24: "The wide range in $EF_{PM}$ values here could be due to the difference in the selection of excavators emission standards"; p13, line 5-8: "As shown in Figure 4, reductions in the measured $EF_{PM}$ between the China II and China IV trucks and between the China III and China IV trucks were 63.5% and 65.6%, indicating that improvement in the emission standards for diesel trucks could significantly reduce PM emissions." Conclusions: "The average $EF_{PM}$ for excavators with different emission standards decreased by 58% from pre-stage 1 to stage 2" and "Moreover, the reductions in $EF_{PM}$ from the China II to the China IV truck and from the China III to the China IV truck were 63.5 and 65.6%, respectively, Indicating that improvements to the emission standards for diesel trucks and excavators have significantly decreased PM emissions." Consistent with what both referees showed concerns for and the fact that the authors also admitted this shortcoming in their study, I believe the conclusions made in various parts of the paper (as highlighted above) are not valid and need to be rephrased.

Response: Thanks for the comments, from which we get to know some confusion has been caused in last run of response, and make several corrections in the revised manuscript.

(1) The sentence "Considering the limit of sample size of our study, it was difficult to calculate the influence of the fuel quality and the emission standards on PM constituents separately." has been revised to "considering the limitation of the experimental design of our study, it was difficult to determine, fuel quality or emission standard, which has the more significant influence on constituents of PM from excavator emissions."

(2) We have extended the discussions about the influences of emission standard and fuel quality on $EF_{PM}$ and rephrased some improper discussions and conclusions in the revised manuscript. The revisions are listed as below.

"Given that there is no government supervision of diesel used for non-road vehicles, the reduction of average $EF_{PM}$ from pre-stage 1 to stage 2 could mainly attribute to

both the different emission standards and diesel quality. As shown in Table S5, the average $EF_{PM}$ from E5 to E6 with the same fuel quality but different emission standards reduced 87.1%. Similarly, $EF_{PM}$ reduced 38.2% from E2 to E1. From which it indicated that emission standards have significant impacts on $EF_{PM}$. Likewise, the average $EF_{PM}$ for E3, E1 and E6 that were under the same emission standard decreased with improvement of fuel quality, suggesting the influence of diesel quality." (Page 13: 21-27).

"The diesel used for trucks was assumed to have identical quality because of strict diesel quality regulations of on-road trucks. Therefore, the reductions of $EF_{PM}$ for different trucks could be mainly attributed to the improvement in the emission standards." (Page 14: 28-30).

"Although the $EF_{PM}$ for excavators and trucks was reduced with the constraint of regulations, the elemental fractions for excavator emissions ranged from 0.49 to 3.03% from pre-stage 1 to stage 2, and the fraction of WSIs for the China IV truck was 6-fold higher than the average value of WSI for all other-levels trucks." (Page 2: 21-25).

"The average EFPM for excavators with different emission standards excavators was decreased by 58% from pre-stage 1 to stage 2. Moreover, the reductions in EFPM from the China II to the China IV truck and from the China III to the China IV truck were 63.5 and 65.6%, indicating that the improvements to of the emission standards and fuel quality for diesel trucks and excavators have significantly effects on the reduction of PM emissions." (Page 23: 9-15).

Table S5 PM Fueled-based emission factors for excavators (mg $kg^{-1}$ fuel)

| ID | Emission standards | Produced year | Sulfur in fuel (ppm) | Idling | Moving | Working | Average |
|----|-----|------|------|------|------|------|------|
| E6 | Stage 2 | 2013 | 200 | 141 | 75 | 97.8 | 96.5 |
| E5 | Pre-stage 1 | 2007 | 200 | 1,336 | / | 603 | 749 |
| E1 | Stage 2 | 2013 | 400 | 75.3 | 88.7 | 340 | 289 |
| E2 | Pre-stage 1 | 2007 | 400 | 845 | 582 | 422 | 468 |
| E3 | Stage 2 | 2012 | 700 | 513 | 331 | 1,214 | 1,047 |
| E4 | Pre-stage 1 | 2004 | 1100 | 559 | 636 | 2,750 | 2,323 |

Comments #2: In response to one of my previous comments (#17) authors mentioned that ambient PM throughout the paper refers to particles smaller than 100 μm. Line 4, page 4, I can't imagine that in a country where dust can also contribute significantly to ambient PM, >90% of PM is from diesel vehicles. What size range (and under what conditions) are these statistics applicable to?

Response: Thanks. This statement reviewer mentioned was inaccurate and the correct one was that "based on the China vehicle environmental management annual report for 2015, 0.56 million tons $PM_{2.5}$ were emitted from on-road mobile sources, of which 90% resulted from on-road diesel vehicle emissions". Note that part of "PM" are not specially defined in the test were mainly referred to ambient fine particle (Dp ≤ 2.5 μm), because almost all of the particles emitted from diesel combustion are fine particles (An et al. 2011). Although we used TSP sampler to collect PM (Dp ≤ 100 μm) in this study, the PM collected in this study was primarily as fine particles with aerodynamic diameter less than 2.5 μm. As shown in Figure R1, ambient fine PM (Dp ≤ 2.5 μm) accounted for around 90% of the PM mass that emitted from excavators in this study. The inaccurate statements have been modified in the revised manuscript (Page 3: 19, 16; Page 4: 14-15; Page 6: 18; Page 8: 20-22).

[Figure]

Figure R1 Average mass fraction of PM under different particle size for excavators

Comments #3: comment #4: >100% recovery for some species are reported. Were the

final numbers adjusted for this recovery rate? (also species that had <100% recovery)?

Response: The final concentrations of organic matters were not corrected with the recovery rate (Page 10: 19).

Comments #4: Comment #10: It's unclear why having used a truck for carrying ironstone would lead to higher Fe, Cu, Zn in the exhaust? Your response suggests these species were not coming from the exhaust, but yet the text suggests otherwise. This makes me wonder then if other measurements are also affected by ambient concentrations. The sections on elemental composition and differences are not well supported in my opinion.

Response: Given that the sampling tube was placed directly on the tailpipe, the previous assumption that samples were affected by ambient particles was improper. Wang et al. (2003) reported that the concentrations of the crustal elements (Fe, Ca, and Mg) account for 50% of the total elements in diesel fuel, which is significantly higher than anthropogenic elements emitted from diesel vehicle engines. Besides, Lin et al. (2015) found that Zn and Cu were originated from lubricating oil, except for brake linings. Therefore, we supposed that diesel and lubricating oil combustion were the main sources of elements exhausted from E4. The detail information could be found in the revised manuscript (Page 16: 21-30; Page 17: 1-3).

Comments #5: Comment #11: why not report trace metal EFs for individual excavators? Having the sum for E1+E6 and the rest doesn't make sense.

Response: Thanks. In last run of response, it was not our purpose that reflect individual excavator, therefore, we just reported the average values of E1+E6 and the rest. Now, Trace mental EFs for individual excavators were given separately in the revised manuscript if the review thought it was necessary. The emission factors of elemental were 4.09, 4.10, 1.71, 8.73, 1.56 and 7.24 mg kg$^{-1}$ fuel for E1, E2, E3, E4, E5 and E6, respectively (Page 17: 5-6).

Comments #6: Comment #16: also define BaP(eq) in line 10, page 11

Response: Thank you for the reminding. The BaP(eq) has been defined in the revised manuscript (Page 12:5-9).

Comments #7: Section 3.3 still needs proof-reading

Response: Thanks. The section 3.3 has been proofed-reading.

Comments #8: Section 3.4: fully write out all the PAHs

Response: We take the reviewer's suggestion, and all the PAHs have been fully written in the revised manuscript (Page 20: 18-19; Page 21: 4, 14, 28).

**Referee #2**

Comments #1: I found a few instances where the language could be improved and I would recommend the authors to go back to their native English speaker to copyedit the manuscript once more.

Response: Thanks for the reviewer's kind suggestion. The manuscript has been reviewed again by our native English speaker.

Comments #2: The response reads very poorly and seems to have been written in a rush (similar to how the manuscript was in the first round). Again, I would recommend the authors to go back to their native English speaker to copyedit the response because the response is published too.

Response: Thanks. The response has been copyedited by our native English speaker.

Comments #3: Add lower bound error bars to Figure 3.

Response: The lower bound error bars have been added in Figure 3 in the revised manuscript (Page 40).

Comments #4: 'Sulfur' misspelled in Figure S5.

Response: The misspelled word has been corrected in the revised Supporting Information (Page 5).

Comments #5: Increase font size for axis labels on Figure 1.

Response: The font size for axis labels on Figure 1 has been increased in the revised manuscript (Page 38).

Comments #6: Emphasize the role of the on-board measurement in this study and state during the comparison to earlier work if the discrepancy in comparison is due to the on-board versus chassis measurement.

Response: Thanks. We have emphasized the role of on-board measurement in the revised manuscript (Page 3: 4-5).

Comments #7: I do not agree with the authors that the quartz fiber filters are equivalent to Teflon filters for gravimetric measurements and that they are insensitive to fiber loss (see comment #7 in response). I would recommend that they provide evidence for that claim.

Response: Thanks. Table R1 listed the results of gravimetric measurements for quartz and Teflon filters in this study. We found that the PM weight from quartz fiber filters were almost equivalent to that from Teflon filters after gravimetric measurements ($Y=0.89X+0.006$; $R^2=0.74$) (as shown in Table R1). Furthermore, there were several errors in some weight of samples using Teflon filters (negative values marked with red in Table R1). Therefore, we chose quartz fibers to conduct gravimetric measurements.

Table R1 Results of gravimetric measurements for quartz and Teflon filters

| ID | Weight (mg) | Volume (m³) | Concentration (mg/m³)[a] |
|----|-------------|-------------|--------------------------|
| Q1[b] | 1.94 | 0.36 | 16.23 |
| T1[c] | 2.28 | 0.40 | 17.10 |
| Q2 | 2.19 | 0.29 | 30.60 |
| T2 | 3.17 | 0.35 | 37.26 |
| Q3 | 1.50 | 0.23 | 16.08 |
| T3 | 1.05 | 0.28 | 9.28 |
| Q4 | 2.40 | 0.39 | 13.57 |
| T4 | 1.28 | 0.31 | 9.14 |
| Q5 | 1.61 | 0.46 | 6.69 |
| T5 | 1.76 | 0.32 | 10.39 |
| Q6 | 6.67 | 0.752 | 14.20 |
| T6 | 2.48 | 0.361 | 10.98 |
| Q7 | 0.62 | 0.12 | 5.00 |
| T7 | 1.38 | 0.24 | 5.75 |
| Q8 | 3.46 | 0.13 | 26.21 |
| T8 | 2.94 | 0.21 | 14.07 |
| Q9 | 0.81 | 0.18 | 4.63 |
| T9 | 0.99 | 0.38 | 2.63 |
| Q10 | 0.63 | 0.16 | 3.87 |
| T10 | 1.25 | 0.36 | 3.49 |
| Q11 | 0.06 | 0.18 | 0.50 |
| T11 | 0.12 | 0.12 | 0.66 |
| Q12 | 0.86 | 0.71 | 1.99 |
| T12 | 1.00 | 0.22 | 2.73 |
| Q13 | 1.90 | 0.61 | 5.29 |
| T13 | 2.22 | 0.50 | 3.56 |
| Q14 | 1.05 | 0.49 | 9.56 |
| T14 | 2.11 | 0.40 | 11.23 |
| Q15 | 0.50 | 0.24 | 12.61 |
| T15 | -1.43 | 0.20 | -1.15 |
| Q16 | 1.19 | 0.40 | 7.33 |
| T16 | -1.17 | 0.35 | -1.34 |
| Q17 | 2.19 | 0.21 | 10.43 |
| T17 | -0.36 | 0.18 | -1.99 |
| Q18 | 2.17 | 0.16 | 13.23 |
| T18 | -0.24 | 0.13 | -1.75 |

a refers to that the concentrations were rectified by dilution ratios;

b refers to quartz filters;

c refers to Teflon filters;

References:

[revised manuscript text omitted]

---

## Author Response (AR3)

Response to editor' Comment on Manuscript: acp-2016-1038

Dear Editor,

We are thankful very much to you for the profound comments. We have made every effort to polish our English. The manuscript was revised by a specialist language service institution (Elsevier Language Editing) and we also asked a native English speaker to take a proof reading of the final version of the revised manuscript.

Best regards,

Dr. Yingjun Chen

[revised manuscript text omitted]